



**Evaluating Model Parameterizations of Submicron Aerosol**
**Scattering and Absorption with In Situ Data from ARCTAS**
**2008**
**M. J. Alvarado[1], C. R. Lonsdale[1], H. L. Macintyre[2,*], H. Bian[3,4], M. Chin[4], D. A.**
**Ridley[5], C. L. Heald[5,6], K. L. Thornhill[7], B. E. Anderson[7], M. J. Cubison[8,**], J. L.**
**Jimenez[8], Y. Kondo[9], L. K. Sahu[9], J. E. Dibb[10], and C. Wang[2]**
[1]{Atmospheric and Environmental Research, Lexington, Massachusetts, USA}
[2]{Center for Global Change Science, Massachusetts Institute of Technology, Cambridge,
Massachusetts, USA}
[3]{Goddard Earth Sciences and Technology Joint Center for Earth Systems Technology,
University of Maryland Baltimore County, Baltimore, Maryland, USA}
[4]{NASA Goddard Space Flight Center, Greenbelt, Maryland, USA}
[5]{Department of Civil and Environmental Engineering, Massachusetts Institute of
Technology, Cambridge, Massachusetts, USA}
[6]{Department of Earth, Atmospheric and Planetary Science, Massachusetts Institute of
Technology, Cambridge, Massachusetts, USA}
[7]{NASA Langley Research Center, Hampton, Virginia, USA}
[8]{Department of Chemistry and Biochemistry, and Cooperative Institute for Research in the
Environmental Sciences, University of Colorado, Boulder, Colorado, USA}
[9]{Department of Earth and Planetary Science, University of Tokyo, Tokyo, Japan}
[10]{Department of Earth Sciences and Institute for the Study of Earth, Oceans, and Space,
University of New Hampshire, Durham, New Hampshire, USA}
[*]{Now at Public Health England, Chilton, Oxfordshire, UK}
[**]{Now at Tofwerk AG, Thun, Switzerland}
Correspondence to: M. J. Alvarado (malvarad@aer.com)



**Abstract**
Accurate modeling of the scattering and absorption of ultraviolet and visible radiation by
aerosols is essential for accurate simulations of atmospheric chemistry and climate. Closure
studies using in situ measurements of aerosol scattering and absorption can be used to
evaluate and improve models of aerosol optical properties without interference from model
errors in aerosol emissions, transport, chemistry, or deposition rates. Here we evaluate the
ability of four externally mixed, fixed size distribution parameterizations used in global
models to simulate submicron aerosol scattering and absorption at three wavelengths using in
situ data gathered during the 2008 Arctic Research of the Composition of the Troposphere
from Aircraft and Satellites (ARCTAS) campaign. The four models are the NASA Global
Modeling Initiative (GMI) Combo model, GEOS-Chem v9-02, the baseline configuration of a
version of GEOS-Chem with online radiative transfer calculations (called GC-RT), and the
Optical Properties of Aerosol and Clouds (OPAC v3.1) package. We also use the ARCTAS
data to perform the first evaluation of the ability of the Aerosol Simulation Program (ASP
v2.1) to simulate submicron aerosol scattering and absorption when in situ data on the aerosol
size distribution is used, and examine the impact of different mixing rules for black carbon
(BC) on the results. We find that the GMI model tends to overestimate submicron scattering
and absorption at shorter wavelengths by 10-23%, and that GMI has smaller absolute mean
biases for submicron absorption than OPAC v3.1, GEOS-Chem v9-02, or GC-RT. However,
the changes to the density and refractive index of BC in GC-RT improve the simulation of
submicron aerosol absorption at all wavelengths relative to GEOS-Chem v9-02. Adding in
situ size distribution information, as in ASP v2.1, improves model performance for scattering
but not for absorption, likely due to the assumption in ASP v2.1 that BC is present at a
constant mass fraction throughout the aerosol size distribution. Using a core-shell mixing
state in ASP overestimates aerosol absorption, especially for the fresh biomass burning
aerosol measured in ARCTAS-B, suggesting the need for time-varying mixing states in future
versions of ASP.





## 1 Introduction

Atmospheric aerosols can both scatter and absorb ultraviolet and visible (UV-VIS) light, thereby altering the actinic flux and the rates of photolytic reactions in the atmosphere (e.g., Michelangeli et al., 1992; He and Carmichael, 1999). The absorption of UV-VIS light by atmospheric aerosols is dominated by light absorbing carbon (LAC) and mineral dust particles (Bian et al., 2003; Martin et al., 2003). Produced by the incomplete combustion of fossil fuels and biomass, LAC has two major forms: "black carbon" or BC, which is primarily composed of soot; and organic aerosols (OA) that strongly absorb UV-VIS light, called "brown carbon" or BrC (Andreae and Gelencsér, 2006). Both forms of LAC can be internally mixed with or coated by less absorbing, more reflective inorganic and organic species, altering their optical properties (e.g., Liao et al., 1999; Yang and Levy, 2004; Lack and Cappa, 2010).

In situ and regional studies of the impact of LAC aerosols on photolysis rates have found that absorbing aerosols can reduce local photolysis rates and OH concentrations by as much as 40% (Tang et al., 2003; Lefer et al., 2003; Alvarado et al., 2009), substantially reducing the net production rate of $O_3$ in urban airsheds (Jacobson, 1998; Li et al., 2005) and biomass burning plumes (Tang et al., 2003; Alvarado et al., 2009), with the magnitude of the impact dependent on the concentrations of $NO_x$ and VOCs (He and Charmichael, 1999; Yang and Levy, 2004). Global modeling studies have found similar impacts of LAC on photolysis rates, OH concentrations, and net $O_3$ production (e.g., Liao et al., 2003). For example, Bian et al. (2003) found that the scattering and absorption of UV-VIS light by aerosols increased global tropospheric mean $O_3$ by ~1 ppbv and decreased OH by 8%. Martin et al. (2003) found that the light absorption by externally mixed black carbon aerosols decreased the modeled photolysis rate of $O_3$ to form $O(^1D)$ by a factor of 2 in biomass burning regions and the Ganges Valley, thus decreasing OH concentrations by as much as 40%. Tie et al. (2005) found 10 to 40% reductions in the formation of $O(^1D)$ by photolysis in Europe, eastern Asia, and the Amazon due to externally mixed anthropogenic and biomass burning aerosols. This caused 5 to 40% reductions in $HO_x$ concentrations along with modest changes to $O_3$ (-4 to +5%).

Furthermore, the scattering and absorption of UV-VIS light by LAC aerosols can lead to a significant climate forcing (the direct effect), but the magnitude of this forcing is uncertain. For example, the review of Bond et al. (2013) estimated the direct radiative forcing (DRF) of atmospheric BC is +0.71 W m$^{-2}$ with 90% uncertainty bounds of (+0.08, +1.27) W m$^{-2}$.




Chung et al. (2012) used data from the ground based Aerosol Robotic Network (AERONET,
Dubovik and King, 2000) to estimate a similar global DRF of $0.65 \pm 0.15$ W m$^{-2}$ from all
LAC aerosols, but in this study brown carbon was estimated to account for ~20% of the total
forcing. However, Wang et al. (2014) used the GC-RT model (Heald et al., 2014) combined
with AERONET data to get a lower DRF estimate of 0.32 W m$^{-2}$ from all LAC aerosols
(uncertainty range 0.04 to 0.50 W m$^{-2}$), with 34% of the forcing coming from BrC.
Accurately accounting for the scattering and absorption of UV-VIS light by LAC aerosols is
thus critical for models of atmospheric composition, air quality, and climate change.
However, in order to reduce computational intensity, most global chemical transport models
(CTMs), such as the Global Modeling Initiative (GMI) Combo model (Duncan et al., 2007) of
the US National Aeronautics and Space Administration (NASA) and GEOS-Chem (Bey et al.,
2001), account for this absorption assuming that all aerosol species are externally mixed (i.e.,
sulfate, sea salt, dust, OA, and black carbon aerosols are not present in the same particle), and
that each of these aerosol types have fixed, prescribed size distributions. These simplifications
can lead to substantial errors in simulating the impact of LAC aerosols on photochemistry, as
these impacts can vary substantially with aerosol size and mixing state. For example, the
studies of Liao et al. (1999) and Yang and Levy (2004) showed that internal mixtures of
sulfate and BC aerosols can cause larger reductions of photolysis rates than external mixtures.
Other theoretical (e.g., Jacobson, 2001) and observational (e.g., Schwarz et al., 2008;
Shiraiwa et al., 2010; Lack et al., 2012) studies suggest that coatings on BC aerosol can
enhance absorption by 30% or more. For example, Kim et al. (2008) showed that accounting
for internally-mixed aerosols and changing aerosol size distributions with time gave a much
smaller total negative TOA forcing ($-0.12$ W m$^{-2}$) of all carbonaceous and sulfate aerosol
compounds compared to the cases using one-moment scheme either excluding or including
internal mixtures ($-0.42$ and $-0.71$ W m$^{-2}$, respectively). However, core-in-shell Mie
calculations carried out by Lack and Cappa (2010) suggested that a black carbon particle
coated with brown carbon can actually absorb less light than a black carbon particle coated in
non-absorbing material, with reductions in absorption of up to 50% relative to clear coatings.
In addition, Cappa et al. (2012) found little (~6%) enhancement of BC absorption by coatings
in California during the US Department of Energy Carbonaceous Aerosols and Radiative
Effects Study (CARES) in June of 2010.
Thus, while the simplifications used in the global CTMs greatly reduce the computational



expense of global studies of the impact of LAC aerosols on photochemistry, it is important to
quantify the errors in the simulation of aerosol scattering and absorption that results from the
assumption of an external mixture and the chosen size distributions for each aerosol type. In
situ closure studies, like the one in this work, allow the accuracy of the aerosol scattering and
absorption calculations in these models to be assessed independently of the potential errors in
other model processes such as the treatment of aerosol emission, secondary organic aerosol
(SOA) formation, and aerosol wet and dry deposition. In these closure studies, ambient
measurements of aerosol mass and composition are used as inputs to the aerosol optical
property routines of the global models, with the model-calculated aerosol optical properties
evaluated using simultaneous in situ measurements of aerosol scattering and absorption. In
addition, more detailed aerosol models that allow for time varying size distributions and more
complicated internal mixtures of aerosol, such as AER's Aerosol Simulation Program (ASP;
Alvarado, 2008; Alvarado and Prinn, 2009; Alvarado et al., 2015) can also be evaluated in
these closure studies to help determine if the errors in the global model routines are primarily
due to their fixed size distributions, assumptions about external mixtures, or their assumptions
about the refractive indices of LAC.
In this study, we evaluate four aerosol optical property parameterizations used in global
models with in situ data on submicron aerosol scattering and absorption at three wavelengths
(450, 550, and 700 nm for scattering, 470, 532, and 660 nm for absorption) gathered during
the 2008 Arctic Research of the Composition of the Troposphere from Aircraft and Satellites
(ARCTAS) campaign. The four parameterizations evaluated are from the Optical Properties
of Aerosol and Clouds (OPAC v3.1; Hess et al., 1998) software package, the GMI Combo
model, GEOS-Chem v9-02 (Bey et al., 2001), and the baseline configuration of a version of
GEOS-Chem with online radiative transfer calculations (called GC-RT; Heald et al., 2014;
Wang et al., 2014). We also use the ARCTAS data to perform the first evaluation of the
aerosol optical property calculations in ASP v2.1, and investigate how the use of in situ size
distribution information and the use of different mixing rules for BC affects the match with
observations.
Section 2 describes the five aerosol optical property models examined in this study, including
the ASP v2.1 model, while Section 3 describes the ARCTAS data used. Section 4 summarizes
the methodology for the closure studies for both the global models (Section 4.1) and ASP
v2.1 (Section 4.2). The results of the global model and ASP closure studies are discussed in





Sections 5 and 6, respectively. The conclusions of the study and recommendations for future
model development are summarized in Section 7.
## 2 Aerosol Optical Property Models
### 2.1 OPAC v3.1
The OPAC package was first described by Hess et al. (1998), and version 3.1 is available
online at http://opac.userweb.mwn.de/radaer/opac-des.html#ftp. OPAC v3.1 includes
microphysical and optical properties of six water clouds, three ice clouds, and 10 aerosol
components, with size distributions and complex refractive indices chosen to represent typical
cases. The optical properties calculated include normalized extinction, scattering, and
absorption coefficients, single scattering albedo, asymmetry parameter, and the phase
function at 61 wavelengths between 250 nm and 40 μm for up to 8 values of relative
humidity. The aerosol components included are water-insoluble aerosols, water-soluble
aerosols, soot, two size modes of sea salt, four size modes of mineral dust, and sulfate
droplets. A given aerosol is then modeled as an external mixture of these ten aerosol
components.
### 2.2 NASA GMI Combo Model
The NASA GMI Combo model is a modular chemical transport model (Duncan et al., 2007;
Strahan et al., 2007; Bian et al., 2009) that includes treatment of both stratospheric and
tropospheric processes. Major atmospheric aerosol components included in the model are
sulfate, black carbon, OA, dust, and sea-salt using either GOCART (Chin et al., 2002, 2009;
Ginoux et al., 2001, 2004) or the University of Michigan/Lawrence Livermore National
Laboratory IMPACT model (Liu et al., 2007).
Within the NASA GMI Coupled model, the impact of aerosols on photolysis rates is
calculated using the FAST-JX model (v6.5). FAST-JX contains lookup tables of the
wavelength dependent extinction efficiencies, single scattering albedos, and phase function
coefficients for 14 aerosol types at 4 wavelengths (300, 400, 600, and 1000 nm) and at 7
values for relative humidity (0%, 50%, 70%, 80%, 90%, 95%, and 99%; see Supplement).
The 14 aerosol types include OA, black carbon, tropospheric, volcanic, and stratospheric
sulfate, two modes of sea salt, and seven modes of mineral dust. The optical properties in
these tables are based on the Mie theory calculations (Mishchenko et al., 2002; Martin et al.,





2003), which were initially performed for GEOS-Chem. The relative humidity dependent
complex indices of refraction and lognormal size distributions are taken from the Global
Aerosol Data Set (GADS) of Köpke et al. (1997), which is in turn based on OPAC v3.1
except that (a) all standard deviations of the modes are set to 2.0 and (b) the dry mode radius
of sulfate used in OPAC v3.1 (0.0695 µm) is reduced to 0.05 µm. FAST-JX then interpolates
the aerosol parameters contained in the look-up tables and calculates the average parameters
for external mixtures of the aerosol types.
## 2.3  GEOS-Chem v9-02
The aerosol optical property parameterizations within GEOS-Chem v9-02 (Bey et al., 2001)
follow a similar approach to the NASA GMI model, but the refractive indices and size
distributions of several aerosol components have been updated based on the work of Wang et
al. (2003a,b), Drury et al. (2010), and Jaegle et al. (2011). Table 1 shows the differences in
the lognormal size distribution parameters, densities, and refractive indices for the aerosol
types examined in this study from OPAC v3.1, GMI, GEOS-Chem v9-02, and the baseline
configuration of GC-RT (discussed below). In general, the geometric standard deviation of
the size distribution $\sigma$ used in GMI was reduced from the values of 2.0 to 1.6 for tropospheric
sulfate, OA, and BC, and to 1.5 for the accumulation mode of sea salt. Furthermore, following
Drury et al. (2010) the assumed geometric mean radius ($r_g$) of OA was increased by a factor
of 3, the mean radius of BC was doubled, and the mean radius of sulfate was increased from
0.05 µm to 0.07 µm. Following Jaegle et al. (2011), the mean radius of the accumulation
mode sea salt was reduced from 0.21 µm to 0.09 µm. The refractive index of tropospheric
sulfate was also updated to reflect that of ammonium sulfate, rather than sulfuric acid
aerosols.
## 2.4  Baseline GC-RT
GC-RT (Heald et al., 2014; Wang et al., 2014) is a configuration of GEOS-Chem that is
coupled with the radiative transfer model RRTMG (Iacono et al., 2008) with modified aerosol
optical properties relative to the standard GEOS-Chem code. Here we test the aerosol optical
properties calculated by the "baseline" configuration of GC-RT described by Wang et al.
(2014). The aerosol optical property calculation in the baseline configuration of GC-RT
differs from GEOS-Chem v9-02 in that the BC density and refractive index are adjusted to the
values recommended by Bond and Bergstrom (2006), which have been found to agree better





with observations (Park et al, 2003; Stier et al, 2007; Kondo et al., 2011). These changes are
shown in Table 1.

### 2.5   ASP v2.1

ASP (Alvarado and Prinn, 2009) simulates the gas-phase, aerosol-phase, and heterogeneous
chemistry of young biomass burning smoke plumes, including the formation of $O_3$ and
secondary inorganic and organic aerosol. ASP is a flexible, sectional size-resolved aerosol
model that includes modules to calculate aerosol thermodynamics, gas-to-aerosol mass
transfer (condensation/evaporation), coagulation of aerosols, and aerosol optical properties.
ASP is generally run as a single box model, but it can be implemented as the chemistry
subroutine of larger Eulerian and Lagrangian chemical transport models (e.g., Alvarado et al.,
2009). ASP has been extensively used to study the chemical and physical transformations of
aerosols within biomass burning smoke plumes and the optical properties of aerosols
(Alvarado and Prinn, 2009; Alvarado et al., 2015) including the first simultaneous simulations
of the fluid dynamics, radiative transfer, gas-phase chemistry, and aerosol-phase chemistry in
a young biomass burning smoke plume (Alvarado et al., 2009). However, the aerosol optical
property routines of ASP have not been previously evaluated with in situ data.
In this study we are using ASP v2.1 (Alvarado et al., 2015). The modules of ASP v2.1 most
relevant to the current study are the modules for aerosol size distribution, thermodynamics,
and optical properties. These modules are described in detail below.

### 2.5.1   ASP Aerosol Size Distribution and Thermodynamics

Aerosols are represented in ASP v2.1 as a single, internally-mixed moving-center sectional
size distribution (Jacobson 1997, 2002, 2005). In this representation, size bin boundaries
remain fixed while the mean particle size within the bin is allowed to change with time, and
each particle in a size bin is assumed to have the same composition. In this study, the aerosol
size distributions were modeled at a high resolution by using 40 size bins, 38 logarithmically
distributed between diameters of 10 nm and 20 μm and two bins for particles smaller than 10
nm or larger than 20 μm. Our tests found that increasing the number of bins used in ASP v2.1
to 100 changed the calculated optical properties by only ~1%. In ASP v2.1, the mass fractions
of different aerosol components are assumed to be independent of aerosol size. This
assumption is likely to be violated for aerosols that contain significant amounts of BC (see
Section 6.2), and is planned to be relaxed in future model development.




The inorganic thermodynamics module in ASP v2.1 includes $H_2O$, $NH_3$, the acids $HNO_3$,
HCl, and $H_2SO_4$, the ions $H^+$, $NH_4^+$, $Na^+$, $K^+$, $Ca^{2+}$, $Mg^{2+}$, $SO_4^{2-}$, $HSO_4^-$, $NO_3^-$, $Cl^-$, $OH^-$, and
their various salts (Alvarado and Prinn, 2009; Alvarado, 2008). Binary and mixed activity
coefficients for the various ion pairs are calculated using the Kusik-Meissner approach (Kusik
and Meissner, 1978) when binary coefficient data are available; otherwise they are
constructed using an appropriate combination of the available binary activity coefficients, as
in Kim et al. (1993a) and Steele (2004). Equilibrium constants for electrolyte and gas-particle
equilibrium reactions, as well as the deliquescence relative humidities (DRHs) of the
electrolytes have been updated to match ISORROPIA II (Fountoukis and Nenes, 2007). The
water associated with the inorganic aerosol is calculated via an iterative solution based on the
Gibbs-Duhem equation (Steele, 2004; Alvarado, 2008). Equilibrium concentrations of the gas
and aerosol species are then calculated using the Mass Flux Iteration (MFI) approach of
Jacobson (2005). This approach to inorganic aerosol thermodynamics compares well with
other inorganic aerosol thermodynamics models such as ISORROPIA (Nenes et al., 1998;
Fountoukis and Nenes, 2007), as shown by Steele (2004) and Alvarado (2008).
The ASP organic aerosol thermodynamics routine is based on the assumption that the organic
aerosol species can partition to both the aqueous phase containing $H_2O$ and the inorganic ions,
and to a hydrophobic organic phase consisting solely of organic compounds (Alvarado,
2008); this approach is based on the one used in the Model to Predict the Multiphase
Partitioning of Organics (MPMPO) of Pun et al. (2002). Partitioning of organics between the
gas and hydrophobic phase is governed by Raoult's law, while the partitioning of organics
into the aqueous phase is governed by Henry's law. Following Pun et al., we assume that (1)
there is no interaction between the aqueous phase inorganic ions and the aqueous phase
organics and (2) the activity coefficient for the organic ions (formed by the dissociation of
organic acids) are equivalent to those of the corresponding molecular solute. Equilibrium
parameters and activity coefficient estimates were taken from available data or estimated
using structure–activity relationships such as the Myrdal and Yalkowsky (1997)
parameterization for organic vapor pressures and the UNIFAC group contribution method
(Fredenslund et al., 1977). These parameters can be uncertain to a factor of 10 or more,
however the surrogate compounds used in ASP in this study to represent water-soluble and -
insoluble organic aerosol (CBIO and POA1, see Section 4.2 below) both have very low vapor
pressures and thus partition almost completely to the aerosol phase, so these uncertainties
have little impact on this study.





For aqueous organic solutes, we have updated the calculation of the associated water content
to use the "kappa" ($\kappa$) parameterization of organic hygroscopicity of Petters and Kreidenweis
(2007). As in Alvarado and Prinn (2009), the water content for the aqueous organic solutes is
then added to the inorganic aerosol water content calculated as described above. In this study,
we assume that $\kappa = 0.04$ for the organic aerosol, corresponding to an O:C ratio of 0.25
(Jimenez et al., 2009). While this value may be highly uncertain, the fact that the aerosol
optical property measurements in this study were generally made at very low relative
humidity (below 20%, see Section 4.1) minimizes the impact of this assumption.
**2.5.2  ASP Aerosol Optical Properties**
As part of this work, we have extensively updated the calculations of aerosol optical
properties within ASP v2.1 beyond those described by Alvarado (2008) and Alvarado and
Prinn (2009). We have implemented spectrally-varying complex refractive indices for
wavelengths between 250-700 nm for five aerosol components ($H_2O$, soot, sulfate, sea salt,
and OA) based on those from OPAC v3.1 (Hess et al., 1998; see also Section 2.1 above).
Similar to the procedure used in the NASA GMI Combo model and GEOS-Chem, we assume
(1) that organics follow the OPAC v3.1 refractive indices of so-called "water-soluble
particulate matter", (2) that all sulfate and nitrate salts follow the OPAC sulfate indices, (3)
that all chloride salts follow the OPAC sea salt indices, and (4) that all BC follows the OPAC
soot indices. The real refractive index of the inorganic aqueous solution (if present) is
calculated using the molar refraction approach of Tang (1997) and Tang et al. (1997).
As in Alvarado and Prinn (2009), we assume here that all particles are spherical for the
purposes of calculating their optical properties. ASP v2.1 has also been updated to include
four mixing rules for the refractive indices of black carbon and the other aerosol components:
(1) a volume-average (VA) dielectric constant mixing rule for all aerosol components; (2) a
core-shell (CS) mixing rule, where a spherical core of BC is surrounded by a spherical shell
of all other aerosol components (with the refractive index of the shell calculated using the
volume-average dielectric constant mixing rule); (3) the Maxwell Garnett (MG) mixing rule,
which assumes that BC is present in randomly distributed inclusions within the (Maxwell
Garnett, 1904); and (4) an external mixture (EXT) of BC and the other aerosols components,
with both sets of particles having the same size distributions but with the relative number of
particles determined by the relative volume concentrations. Mie calculations of aerosol optical





properties for each bin of the size distribution are performed within ASP using the publicly
available program DMiLay, which is based on the work of Toon and Ackerman (1981).
**3   ARCTAS Data**
The objective of the NASA ARCTAS campaign (Jacob et al., 2010) conducted in April and
June-July 2008 was to better understand the factors driving current changes in Arctic
atmospheric composition and climate. It used chemical and radiative measurements from
three research aircraft (DC-8, P-3, B-200) to interpret and augment the continuous
observations of Arctic atmospheric composition from satellites. The aircraft were based in
Alaska in April (ARCTAS-A) and in western Canada in June-July (ARCTAS-B). The focus
of ARCTAS-A was to examine the long-range transport of anthropogenic pollution to the
Arctic, while ARCTAS-B was more focused on the impacts of boreal forest fires on regional
and global atmospheric composition. The summer ARCTAS-B deployment was preceded by
one week of flights over California sponsored by the California Air Resources Board (CARB)
to address regional issues of air quality and climate forcing.
Here we use data from the DC-8 aircraft during all three phases of ARCTAS, as described in
detail below, to evaluate the aerosol optical property models. All analyses in this study used
the "merged" data set averaged to the 10s time resolution of the Ultra-High Sensitivity
Aerosol Spectrometer (UHSAS).
**3.1   Aerosol Mass Concentrations and Composition**
On the NASA DC8, submicron black carbon mass was measured with the University of
Tokyo Single Particle Soot Photometer (SP2; Moteki and Kondo, 2007, 2008). The SP2
measures BC volume for particles with volume equivalent diameters between 90 nm and 1.0
$\mu$m. The measured BC volume is then converted to BC mass using an assumed density of 1.8
g/cm$^3$. The uncertainty in the BC mass measurements is ±30%.
An Aerodyne high-resolution, time-of-flight aerosol mass spectrometer operated by the
University of Colorado, Boulder (HR-ToF-AMS, hereafter AMS; DeCarlo et al., 2008;
Cubison et al., 2011) was used to measure ammonium, chloride, nitrate, sulfate, and organic
aerosol mass concentrations. The AMS primarily samples submicron aerosols, with 0%
transmission for vacuum aerodynamic diameters below 35 nm and an approximate $PM_1$ size
cut in vacuum aerodynamic diameter (DeCarlo et al., 2004; Canagaratna et al., 2007). The





uncertainty of the AMS mass concentrations measurements (2σ) is ±34% for the inorganics
and ±38% for the organics.
In addition, data on the concentration of water-soluble organic carbon (WSOC) in submicron
aerosol is provided by the Particle In Liquid Sampler (PILS-WSOC) of the Georgia Institute
of Technology (Sullivan et al., 2006), with an uncertainty of ±45%. The measured WSOC
was converted to total organic mass using a factor of 1.6. This value is uncertain to at least
±0.4, but as our total OA concentration is determined by the AMS and the relative humidites
of the optical property measurements were low, this assumption has little impact on our
results. The PILS-WSOC data is used in the ASP closure study to separate water-soluble and
water-insoluble organic aerosol (see Section 4.2 below).
The AMS data were also supplemented with measurements of additional inorganic cations
from the University of New Hampshire Soluble Acidic Gases and Aerosol (UNH SAGA,
Dibb et al., 2003) instrument. SAGA collects non-size selected ("bulk"), isokinetically-
sampled aerosols onto a teflon filter. The ions are then extracted off the filter with deionized
water and analyzed via ion chromatography. In addition to the ions measured by the AMS,
SAGA provides measurements of the refractory cations sodium (±0.1 ug/m$^3$ at 1013 hPa and
273.15 K), potassium (±0.017 ug/m$^3$), magnesium (±0.011 ug/m$^3$), and calcium (±0.018
ug/m$^3$). In order to convert the SAGA bulk measurements of these ions into submicron
concentrations consistent with the AMS time-resolution, we use the bulk SAGA data to
determine a bulk ratio of these refractory cations to aerosol sulfate, and to combine these
ratios with the AMS measured sulfate concentrations to estimate the submicron mass
concentrations of the refractory cations.
To test this procedure, we compare AMS measured submicron nitrate, ammonium, and
chloride mass concentrations versus the concentrations estimated with the above SAGA-based
procedure. The match is very good for nitrate and ammonium (not shown), but submicron
chloride (not shown) is larger by the SAGA based procedure, as expected since SAGA is
sensitive to refractory chlorides such as NaCl and the AMS is not. However, as we expect the
aerosol in the ARCTAS campaign to be dominated by OA and BC aerosols, this should have
little impact on our closure study results.



## 3.2  Aerosol Size Distribution
In this study, we use the in situ measurements of dry aerosol size distribution provided by the
instruments of the NASA Langley Aerosol Research Group (LARGE; Anderson et al., 1998).
Specifically, we use the dry aerosol size distribution data from the TSI Scanning Mobility
Particle Sizer (SMPS), the Droplet Measurement Technologies (DMT) Ultra-High Sensitivity
Aerosol Spectrometer (UHSAS), and the TSI Aerodynamic Particle Sizer (APS) Model 3321.
The UHSAS is our primary source of size distribution information as the size range measured
by the UHSAS (optical particle diameters between 60 nm to 1000 nm) measures the particles
most likely to affect optical properties in the UV-VIS. The UHSAS has 99 bins geometrically
distributed in this size range, and gathers data every 10 s. The estimated precision of the
UHSAS is 5% in the particle size, and 20% in the particle number concentrations in each bin.
The TSI SMPS measures dry aerosol size distributions in 54 size bins with geometric
diameters between 8.8 nm and 399.7 nm. The time resolution is slower than the UHSAS
(105s for the SMPS versus 10 s for the UHSAS), and so care has to be taken in matching
SMPS size distributions with the UHSAS, as described in Section 4.2. The estimated
precision of the SMPS is 5% in the particle size, and 25% in the particle number
concentrations in each bin.
The TSI APS measures dry aerosol size distributions in 13 size bins with aerodynamic
diameters between 0.583 μm and 7.75 nm. The time resolution is the same as the UHSAS, but
the measured diameters are different (optical diameter for the UHSAS, aerodynamic diameter
for the APS). Converting continuum-regime aerodynamic diameter $D_{aero}$ to geometric
diameter $D_{geo}$ is done through the equation:
$$D_{geo} = D_{aero} \sqrt{\frac{X}{\rho}}. \qquad (3)$$
where $\rho$ is the particle density and $X$ is the "dynamic shape factor" that accounts for the non-
sphericity of the particles (for spheres, $X = 1.0$, otherwise $X > 1$). The estimated precision of
the APS is 10% in the particle size, and 20% in the particle number concentrations in each
bin.




### 3.3  Aerosol Optical Properties
In this study, we use the in situ measurements of dry aerosol scattering and absorption
provided by the LARGE suite of instruments. During ARCTAS, LARGE measured dry total
aerosol scattering and hemispherical backscattering coefficients at three wavelengths (450
nm, 550 nm, and 700 nm) using a TSI model 3563 nephelometer with an estimated precision
of 0.5 Mm$^{-1}$. These total scattering coefficients were then corrected for truncation errors using
the procedure described by Anderson and Ogren (1998). A Radiance Research (RR)
nephelometer with a 1 μm cut cyclone measured the scattering of submicron aerosols at 532
nm. This data allowed an estimate of the submicron scattering at 450nm, 550 nm, and 700 nm
by comparison of the two nephelometers when they were sampling mainly submicron
particles (i.e., a fine mode fraction > 0.6).
Dry total and submicron absorption was also measured at three wavelengths (467 nm, 532
nm, and 660 nm) using a RR Particle Soot Absorption Photometer (PSAP) with an estimated
precision of 0.2 Mm$^{-1}$. These filter-based absorption measurements were corrected to in situ
values using two methods: one from Virkkula (2010) and one from Lack et al. (2008). These
two corrections can differ by about 20-30%, with Virkkula (2010) giving lower aerosol
absorption. Most of our analysis is based on the correction of Lack et al. (2008), but we
discuss the sensitivity of our conclusions to the choice of correction as well.
## 4  Closure Study Methodology
### 4.1  Fixed Size Distribution Parameterizations
As OPAC, GMI, GEOS-Chem, and GC-RT all share a common heritage and features (i.e.,
external mixtures of fixed size distributions of the various aerosol components) our closure
study methodology for all four parameterizations is also similar. The general procedure is
shown in Figure 1a. The first step is to assign the measured aerosol mass concentrations to the
different aerosol types. In this study we focus on tropospheric, submicron aerosol, as detailed
composition data is available from the AMS and SP2 for this size range, and thus we exclude
the mineral dust, stratospheric sulfate, volcanic sulfate, and coarse mode sea salt aerosol
types. The SP2-measured submicron mass of BC is assigned to the BC (for GMI, GEOS-
Chem, and GC-RT) or SOOT (for OPAC) aerosol types. The AMS-measured submicron OA
mass is assigned to the OA (for GMI, GEOS-Chem, and GC-RT) or WASO (for OPAC)
aerosol types. For the inorganic species measured by the AMS and SAGA, we calculate



"equivalent electrolytes" consistent with the measured and estimated submicron ion
concentrations (see Equation 17.72 of Jacobson, 2005). The sulfate- and nitrate-containing
electrolytes are then assigned to the tropospheric sulfate (for GMI, GEOS-Chem, and GC-RT)
and SUSO (for OPAC) aerosol types, while the chloride-containing electrolytes are assigned
to the accumulation mode sea salt type.
The second step involves determining the submicron scattering coefficient, absorption
coefficient, single scattering albedo (SSA), and asymmetry parameter for each aerosol type at
the measured wavelengths and relative humidities (RHs). This is done through linear
interpolation of the values present in the look-up tables for each aerosol parameterization. As
the LARGE instruments measure dry optical properties, the RH used in the interpolation
should not be the ambient RH, but instead is the RH in the inlet prior to the measurement.
Here, we used the measured inlet RH in all comparisons. This "dry" RH is generally near 0%,
but can get as high as 20%.
For GMI, GEOS-Chem, and GC-RT, the tabulated properties include the extinction efficiency
($Q_{ext}$), effective radius ($r_{eff}$), and SSA ($\omega$) for each aerosol type $j$. After correction for
wavelength and RH as described above, the extinction coefficient ($k_{ext,j}$) for each aerosol type
is calculated from these properties by the equation:
$$k_{ext,j} = \frac{3}{4} \frac{Q_{ext,j}}{r_{eff,j}} \frac{m_j}{\rho_j}.$$   (4)
where $m_j$ is the mass concentration for each aerosol type ($\mu g/m^3$, corrected to ambient
temperature and pressure) and $\rho_j$ is the particle density. The scattering and absorption
coefficients are then calculated as $k_{scat,j} = \omega_j k_{ext,j}$ and $k_{abs,j} = (1 - \omega_j) k_{ext,j}$. The $k_{ext}$, $k_{scat}$, $k_{abs}$,
and $\omega$ for each aerosol type are then combined together to give the model estimate of the
optical properties for the submicron aerosol mixture. For $k_{ext}$, $k_{scat}$, and $k_{abs}$ this is a simple
sum, e.g. $k_{scat} = \sum_j k_{scat,j}$, and $\omega$ is the ratio of $k_{scat}$ to $k_{ext}$.
For OPAC, the tabulated properties include values of $k_{ext,j}$, $k_{scat,j}$, $k_{abs,j}$, and $\omega_j$, with $k_{ext,j}$, $k_{scat,j}$,
and $k_{abs,j}$ normalized to an assumed particle number concentration of 1 $cm^{-3}$. These normalized
values are multiplied by the ratio of the measured mass concentration $m_j$ to the assumed mass
concentration for 1 particle $cm^{-3}$ for each aerosol type. These properties are then corrected for



wavelength and RH and combined together as described above for the GMI and GEOS-Chem
parameterizations.

### 4.2   ASP v2.1

The ASP v2.1 model closure studies (see Figure 1b) differ from the other closure studies
mainly in the use of the data on the in situ dry aerosol size distribution from the LARGE
instrument suite. As noted in Section 3.2 above, this data comes from three different
instruments (the SMPS, UHSAS, and APS) with different time resolutions and measuring
techniques. Thus combining these observations into a consistent picture of the size
distribution is not a straightforward task. Our approach uses the UHSAS observations as the
core of our size distribution estimate, as the submicron aerosol optical properties of interest
here are likely most sensitive to aerosol within the size range of the UHSAS (60 nm – 1000
nm). We start by creating a "combined" size distribution with the same size resolution as the
UHSAS observations, but with an expanded range (i.e., 246 bins with optical diameters
between 8.8 nm and 10 μm). For size bins with diameters between 60 nm and 850 nm, the
UHSAS data is used directly. For size bins below 60 nm, SMPS data (interpolated to the
UHSAS size resolution) is used. As the SMPS has a lower time resolution than the UHSAS,
we scale the SMPS data to match the UHSAS data in the size range 60 nm to 100 nm – the
scale factor is the slope of the linear regression of the (interpolated) SMPS and UHSAS data
in this size range. For size bins larger than 850 nm, the size distribution is based on the APS
data, with the conversion factor between aerodynamic and geometric diameter assumed to be
0.8. This value is consistent with the density and shape factors of urban aerosols and solid
ammonium sulfate (Reid et al., 2006), and for a spherical particle is equivalent to a density of
1.56 g cm$^{-3}$. The corrected APS data is used to define a power law that describes how the size
distribution decays at optical diameters larger than 850 nm, and this power law is used to
extrapolate the UHSAS data for diameters larger than 850 nm.
As ASP v2.1 requires that the dry aerosol size distributions be input as a sum of lognormal
modes, the "combined" size distribution described above is fit to three lognormal modes (see
Equations 13.18 and 13.20 from Jacobson, 2005). The fitting boundaries for the three modes
are fixed at 8 – 80 nm, 80 – 400 nm, and 400 nm – 10 μm, as these boundaries coincide with
minima in the ARCTAS size distribution data.




The submicron aerosol mass concentrations of BC, OA, and equivalent electrolytes were
calculated as described in Section 4.1. OA was assumed to be fairly involatile and was
assigned to the species CBIO (if water-soluble) and POA1 (if not water soluble; Alvarado,
2008). These mass concentrations define the relative mass composition (i.e. mass fractions) in
the ASP modes. This aerosol composition was assumed to be the same for all three modes
input to ASP v2.1 – while the AMS can be used to get size-resolved composition, the
averaging times required for this data are large (about an hour), and thus are not useful for
comparison to the 10 s resolution optical property data.
We then used ASP v2.1 to calculate total and submicron $k_{scat}$, $k_{abs}$, and $\omega$ for wavelengths
between 250 nm and 700 nm (at 1 nm resolution) using each of the four mixing rules
described above: volume-averaged, core-shell, Maxwell-Garnett, and external mixture. These
were compared with the in situ measurements of optical properties from the LARGE
instruments.
## 5  Fixed Size Distribution Parameterization Results
### 5.1  Scattering
Figure 2 shows a scatterplot of the measured submicron scattering coefficient at 550 nm
versus the value calculated using the optical property tables of the GMI Combo model. The
slope and correlation coefficient ($r^2$) of a linear fit to the data from the entire ARCTAS
campaign are used in evaluating the models; these values are summarized in Table 2. We see
that all four parameterizations explain 70-74% of the variability (e.g., $r^2$ = 0.70-0.74) in the
observed submicron scattering at all three wavelengths, except for the GMI model at 700 nm,
where only 58% of the variability is explained. The slopes of the linear fits are between 0.89
and 1.08 for the 450 and 550 nm channels, but the 700 nm channel shows more variability,
with a slope of 1.19 for OPAC v3.1 and the slopes for the other models between 0.63-0.68.
However, Figure 2 shows that there can be substantial differences in the results for the
different phases of the ARCTAS campaign. The parameterizations generally work best for the
ARCTAS-B campaign, which sampled several fresh biomass-burning plumes and thus likely
had more externally mixed aerosol samples than the other two phases that sampled more aged
pollution. For ARCTAS-B the $r^2$ values were ~0.75, with slopes between 0.99 and 1.15. In
contrast, the models generally overestimate the relatively smaller scattering coefficients of the
aged arctic pollution sampled during the ARCTAS-A campaign, with $r^2$ values of ~0.63 and





slopes between 1.5 (OPAC v3.1) and 2.0 (GEOS-Chem v9-02). The ARCTAS-CARB phase
shows a clear bifurcation, with some samples overestimated by a factor of 2 or more and
some underestimated by similar factors, suggesting two distinct types of aerosols were
sampled in this phase. This leads to poor $r^2$ values for this phase (0.25-0.39) and, as the
largest values are generally underestimated, slopes between 0.40 (OPAC v3.1) and 0.70
(GEOS-Chem v9-02).
We also examined the distribution of the errors (modeled value – measured value) of the
submicron scattering coefficient, as shown in Figure 3 for 550 nm and the GMI model. To
reduce the impact of the large dynamic range of the measured scattering coefficients on our
analysis, we examined the errors in the logarithm (base 10) of the scattering coefficients,
which is equivalent to the logarithm of the ratio of the modeled to measured value.  The mean
($\mu$) and standard deviation ($\sigma$) of these error distributions are also summarized in Table 2. We
prefer these metrics over mean normalized bias (MNB), as with the logarithmic (geometric)
approach an overestimate of a factor of 2 and an underestimate of a factor of 2 average out to
no mean error, while the MNB of these two observations would be 25% due to the asymmetry
of overestimates and underestimates when expressed as percentages. However, the use of
MNB instead of $\mu$ does not substantially alter the conclusions of our study, and values for
MNB are also discussed below. The spread of the errors is very similar for all models and
wavelengths, with $\sigma$ of ~0.25, which is equivalent to a standard deviation of a factor of 1.8
about the mean. The histograms of the errors show little skew to either side of the mean value.
The models give a positive bias at 450 and 550 nm, with the GMI model having the lowest
mean bias in these channels ($\mu = 0.06$ and $0.09$, respectively, equivalent to a geometric mean
overestimate of 15% and 23%, and an MNB of 35% and 46%). As the 450 nm channel is
closest to the UV wavelengths important in photolysis, we would thus expect the GMI model
to perform best in modeling the impact of aerosols on photolysis rates. GEOS-Chem v9-02
has a slightly smaller negative bias (-0.04, equivalent to a geometric mean underestimate of
9% and an MNB of 8.5%) than GMI ($\mu = -0.05$, geometric mean underestimate of 11%, MNB
of 16%) in the 700 nm channel. The results for GC-RT are similar to GEOS-Chem v9-02 at
450 and 550 nm, but the negative bias at 700 nm is about twice as large on average in GC-RT
as it is in GMI or GEOS-Chem v9-02.



## 5.2 Absorption and SSA
Figure 4 shows the shows a scatterplot of the measured submicron absorption coefficient at
532 nm versus the value calculated using the optical property tables of the GMI model. The
PSAP measurements have been corrected using the approaches of Lack et al. (2008) (Figure
4a) and Virkkula (2010) (Figure 4b). As stated in Section 3.3 above, the Virkkula (2010)
correction generally gives 20-30% lower aerosol absorption coefficients than the Lack et al.
(2008) correction. As the models tended to overestimate aerosol absorption using both
corrections, we discuss our results relative to the Lack et al. (2008) corrected values. Results
for all model-wavelength combinations using the Lack et al. (2008) correction are
summarized in Table 3. Unlike for scattering, the absorption coefficient slopes and
correlations are fairly consistent between the ARCTAS-B and ARCTAS-CARB phases of the
campaign, but the ARCTAS-A phase shows larger model overestimates of aerosol absorption
for the aged Arctic pollution sampled in that campaign. The global model parameterizations
can explain 65-72% of the observed variability, comparable to but a little worse than their
performance for scattering (see Section 5.1), with slopes between 0.75 (GMI, 660 nm) and
1.21 (GEOS-Chem v9-02, 532 nm).
Figure 5 shows the histogram of the errors in the logarithm of the submicron aerosol
absorption coefficient for GMI at 532 nm, while the mean and standard deviation for all
model-measurement combinations are summarized in Table 3. The spread of the errors
(measured by the standard deviation $\sigma$) is between 0.24-0.29 for all model-wavelength
combinations, giving a standard error of a factor of ~1.7 to 2.0 around the mean bias. We can
see that while all the models show a positive mean bias for aerosol absorption at all
wavelengths, the GMI model has the smallest mean bias at all wavelengths, with a maximum
bias at 532 nm ($\mu = 0.06$, or a 15% geometric mean overestimate, and an MNB of 39%).
Similar results hold when the Virkkula (2010) correction is used, but the geometric mean
overestimate for GMI at 532 nm increases to 55%. Thus while we can conclude the GMI
parameterization performs the best for submicron aerosol absorption of the global model
parameterizations evaluated in this study, we can only conclude that its geometric mean error
is within the range of 0-55%, depending on wavelength and PSAP correction. Table 3 also
shows that the mean overestimate in aerosol absorption in GEOS-Chem v9-02 has been
substantially reduced, but not eliminated, by the improved values for BC density and
refractive index in GC-RT (e.g., from $\mu = 0.27$ and MNB of 120% in GEOS-Chem v9-02 to $\mu$



= 0.22 and MNB of 95% in GC-RT at 532 nm), and the correlation coefficients are slightly
improved as well.
Figure 6 shows the results for Single Scattering Albedo (SSA) for the GMI model at 550 nm.
The Lack et al. (2008) measured absorptions at 532 nm and 660 nm were used to derive an
absorption Angstrom exponent that was then used to estimate the observed absorption at 550
nm. As expected, since both the GMI scattering and absorption comparisons showed small
positive biases at this wavelength (see Tables 2 and 3), the GMI calculation of SSA is
relatively unbiased, as shown in Table 4. However, the spread of the errors is large ($\sigma = 0.05$),
and the correlation between the modeled and measured values is poor ($r^2 = 0.06$). We
explored whether averaging the observations at 1 and 5 minute intervals would reduce the
spread in the SSA errors and improve the correlation. However, the spread of errors only
decreased to $\sigma = 0.03$ when the data is averaged to 5 minute intervals, and the correlation
coefficient only increased to $r^2 = 0.22$. Thus, while the GMI model gives reasonable mean
values for SSA, the calculated value tends to be significantly wrong for any given data point.
In addition, the GEOS-Chem v9-02 and GC-RT SSA predictions show similar biases and
spread of errors, with GC-RT performing slightly better than the other models at 450 nm.
**6 ASP v2.1 Results**
**6.1 Impact of Size Distribution Data on Aerosol Scattering**
As expected, when ASP v2.1 is given aerosol size distribution data from the LARGE
instrument suite, it does a substantially better job of modeling the observed aerosol scattering
than the global model parameterizations discussed in Section 5.1. Figure 7 shows the
scatterplot and histogram of the errors in submicron aerosol scattering for ASP v2.1 at 550
nm. The results for a core-shell (CS) BC mixing state are shown, but the results for all other
mixing states are similar, as shown in Table 5. Note that there are far fewer data points in
Figure 7a than in Figure 2 for the GMI model (1,771 versus 10,629). This is because the ASP
v2.1 closure requires all three LARGE size distribution instruments to be working at the same
time as the AMS, SP2, and other composition instruments, which reduces the amount of
available data. With the size distribution data, ASP v2.1 with the core-shell mixing state is
able to explain 88-89% of the observed variability in aerosol scattering, with linear regression
slopes of 0.99, 1.00, and 1.07 at 450, 550, and 700 nm, respectively. The maximum mean
(logarithmic) bias is $\mu = -0.03$ (equivalent to a mean underestimate of 7%), and the standard





deviation of the logarithmic errors ($\sigma$) is only 0.17, equivalent to a factor of 1.5. Given that
the size distribution data is itself uncertain to 20% before the instruments are stitched
together, this is remarkably good model performance. Together this implies that ASP v2.1 is
able to model more than 90% of the scattering data points to within a factor of 2.

## 6.2   Impact of Size Distribution Data and Black Carbon Mixing State on Aerosol Absorption

In contrast to the results for scattering, ASP v2.1, with aerosol size distribution data from the
LARGE instrument suite, has difficulty reproducing the observations for aerosol absorption.
Figure 8a shows a scatterplot of the measured submicron absorption coefficient versus the
value calculated by ASP v2.1 using a CS mixing state. While overall slope (0.93 ± 0.12) is
reasonable, there are clear problems in the simulation of the absorption observations from
ARCTAS-B and CARB, leading to a poorer correlation coefficient ($r^2 = 0.44$) and a larger
spread in the errors ($\sigma = 0.32$, see Figure 8b) than was seen for the global parameterizations
using fixed size distributions and external mixtures.
Table 6 summarizes the submicron absorption results averaged over all ARCTAS phases for
the different BC mixing states that can be used in ASP v2.1. The relatively poor correlation
and wide spread of errors is consistent across mixing states. As expected, the assumption of
an external mixture (EXT) results in the lowest modeled absorption, significantly
underestimating absorption at 470 nm and 532 nm, but giving very little bias at 660 nm. In
contrast, the internal mixtures (CS, MG, and VA) all overestimate absorption, but show much
less dependence of this bias on wavelength. CS and VA both give regression slopes near 1,
but the VA mixing state shows a larger overestimate of absorption than the CS mixing state,
while the MG mixing state has a lower positive bias than CS.
However, the results vary significantly between the different phases of ARCTAS. For
example, Figure 9 shows the histograms of the ASP errors when the CS mixing state is used
separated for the three campaigns. We can see that both the mean bias and the spread of the
errors vary significantly between the campaigns. For example, ASP with CS appears
relatively unbiased for ARCTAS-CARB ($\mu = -0.02$) but the spread of the errors is large ($\sigma = 0.36$).
0.36). The results for ARCTAS-A show a small positive bias ($\mu = 0.07$) similar to the overall
GMI results ($\mu = 0.06$, see Table 3), but with a small spread in the errors ($\sigma = 0.18$). In



contrast, ASP with CS substantially overestimates absorption during ARCTAS-B by an
average factor of 2 ($\mu = 0.32$), but again shows a relatively small spread of errors ($\sigma = 0.23$).
These differences between the campaigns make sense when we consider the types of pollution
sampled during each campaign. ARCTAS-A sampled primarily aged Arctic haze particles,
and so the BC in these particles would be expected to be internally mixed. In contrast,
ARCTAS-B sampled substantial amounts of fresh biomass burning smoke, where the BC
would be expected to be externally mixed and thus have lower absorption per mass of BC
than would be calculated by the core-shell assumption. Finally, ARCTAS-CARB sampled a
mixture of anthropogenic pollution and biomass burning smoke from a variety of sources.
These aerosols are likely in a variety of mixing states and have a variety of size distributions
of BC particles, and thus ASP CS would be expected to show the large spread of errors seen.
These results point to the need for further development of the ASP model to allow for time-
varying mixing states and to allow the BC size distribution to vary independently of the
overall size distribution.
In order to examine the benefit that including a time-varying mixing state for BC in ASP
could bring, we examined a third "variable" mixing state case where CS was used for
ARCTAS-A and ARCTAS-CARB while EXT was used for ARCTAS-B. The results are
shown in Figure 10 and Table 6. This variable mixing state generally shows lower mean
positive biases than any of the internally-mixed states (CS, VA, MG) while showing a slightly
smaller spread in the errors than any of the constant mixing state cases ($\sigma$ of 0.28-0.29 versus
0.30-0.32), more consistent with the GMI results seen in Table 3. However, the correlation
coefficient is still very poor ($r^2 = 0.44$-0.45), suggesting that the assumption in ASP v2.1 of a
constant mixing ratio of BC throughout the overall size distribution can lead to errors in
submicron aerosol absorption as large as those seen in the externally-mixed, fixed-size
distribution global models.
Table 7 summarizes the results for SSA for ASP v2.1 using different mixing states. When
assuming internal mixtures, ASP tends to underestimate the SSA by an average of 0.01 to
0.04, while assuming external mixtures gives fairly unbiased results (-0.01 to 0.01). We see
that the "variable" mixing state gives small negative biases similar to the results from the MG
mixing state, but has a higher correlation coefficient ($r^2$ of 0.30 at 532 nm, compared to 0.20
for MG). The SSA correlation coefficients for ASP v2.1 for all mixing states are generally
higher than those for GMI or GEOS-Chem v9-02.



## 7   Conclusions

We performed a closure study using in situ observations of submicron aerosol concentration, composition, size distribution, scattering, and absorption from the NASA ARCTAS campaign to evaluate the modeling of submicron aerosol scattering and absorption in four global parameterizations (those used in the GMI Combo model, OPAC v3.1, GEOS-Chem v9-02, and the baseline configuration of GC-RT) as well as the smoke plume chemistry model ASP v2.1. Our closure study allowed for the evaluation of the predictions of aerosol scattering and absorption by these models without the complications associated with different treatments of aerosol emissions, transport, chemistry, and deposition. We find that the GMI model has smaller mean biases in predicting submicron aerosol scattering and absorption than OPAC v3.1, GEOS-Chem v9-02, or the baseline GC-RT. On average, GMI overestimates submicron aerosol scattering during ARCTAS by 15% ($1\sigma$ range -34% to 100%) at 450 nm and 23% (-29% to 114%) at 550 nm, while it underestimates scattering at 700 nm by -11% (-53% to 66%). When the Lack et al. (2008) correction is applied to the ARCTAS PSAP observations, GMI overestimates submicron aerosol absorption by 10% ($1\sigma$ range -41% to 104%) at 470 nm, by 15% (-38% to 114%) at 532 nm, and by 0% (-42% to 74%) at 660 nm. On average GMI slightly overestimates mean submicron SSA during ARCTAS ($0.01 \pm 0.05$ at 470 and 532 nm, $0.02 \pm 0.07$ at 660 nm) while GEOS-Chem v9-02 slightly underestimates it ($-0.01 \pm 0.05$ at 470 and 532 nm, $-0.02 \pm 0.06$ at 660 nm), but the correlation is very poor for all models, suggesting that while the mean is reasonable the models tend to have little skill predicting individual data points. We also find that the changes to the treatment of BC in the baseline configuration of GC-RT reduce the positive bias in modeled absorption relative to that in GEOS-Chem v9-02 (e.g., from a mean overestimate of 86% in GEOS-Chem v9-02 to a mean overestimate of 66% at 532 nm).

The use of in situ size distribution information allows ASP v2.1 to accurately simulate submicron aerosol scattering with a high correlation ($r^2 = 0.88$-$0.89$) and very little spread in the error distribution compared to the GMI model. When a core-shell (CS) BC mixing state is used, ASP v2.1 underestimates aerosol scattering during ARCTAS by 7% ($1\sigma$ range -37% to 38%) at 450 and 550 nm and 2% (-34% to 45%) at 700 nm on average. However, the ASP v2.1 results for submicron aerosol absorption show a substantially lower correlation ($r^2 = 0.44$-$0.50$) likely due to the assumption in ASP v2.1 of a constant mass fraction of BC throughout the size distribution. When a CS mixing state is used, ASP v2.1 overestimates




submicron aerosol absorption by 29 to 35% with a weak dependence on wavelength, while
ASP v2.1 with an external (EXT) mixture in ASP tends to underestimate aerosol absorption,
with the average errors showing a strong dependence on wavelength (-21% at 470 nm, -11%
at 532 nm, and 0% at 660 nm). Examination of the distribution of errors for each phase of the
ARCTAS campaign suggests that an external mixture is best for the fresh smoke observations
in ARCTAS-B, while an internally mixed core-shell approach is better for the aged Arctic
haze in ARCTAS-A and the anthropogenic pollution in ARCTAS-CARB. Using this
"variable" mixing state in ASP v2.1 leads to an average overestimate of aerosol absorption of
10% ($1\sigma$ range -42% to 109%) at 470 nm, 17% (-38% to 124%) at 532 nm, and 23% (-37% to
140%) at 660 nm.
These results suggest that the GMI model does a reasonable job modeling submicron aerosol
scattering and absorption, and that future refinements to the GMI approach should focus on
improvements that, on average, reduce scattering and absorption in the 550/532 nm and
450/470 nm bands. For GEOS-Chem, adopting the baseline GC-RT BC parameters from
Wang et al. (2014) for the standard GEOS-Chem model would substantially improve the
ability of the model to simulate aerosol absorption. However, further refinements to the
treatment of BC and OA absorption are needed to reduce the positive bias that remains in GC-
RT, such as the potential of BrC absorption to decrease with atmospheric age (e.g., Forrester
et al., 2015). For ASP v2.1, the results show that future model development should focus on
improvements to the simulation of submicron aerosol absorption by adding routines that allow
for a more complete description of aerosol mixing state (such as in the PartMC-MOSAIC
model, Tian et al., 2014) and adding the ability for the BC mass fraction to vary with aerosol
size. In addition, similar closure studies should be performed with data from other recent field
campaigns, such as NASA Studies of Emissions and Atmospheric Composition, Clouds and
Climate Coupling by Regional Surveys (SEAC[4]RS; Toon, 2013) and the US Department of
Energy Biomass Burning Observation Project (BBOP; Kleinman et al., 2014) campaign, to
investigate how the biases in the aerosol optical property models vary with location and
pollution source.
**Acknowledgements**
The authors thank the other members of the ARCTAS Science Team. We also thank Prof.
Rodney Weber of the Georgia Institute of Technology for the use of his PILS data, as well as
Prof. Christopher Cappa of the University of California - Davis and Dr. Manvendra Dubey of



Los Alamos National Laboratory for their helpful comments. This analysis and associated
updates to the ASP model were funded under NASA Grant NNX11AN72G to MJA, CRL,
HLM, HB, MC, and CW, as well as NSF Grant AGS-1144165 to MJA and CRL. DAR and
CLH were partially supported by NASA grant NN14AP38G. JLJ was partially supported by
NASA NNX15AT96G and NNX15AH33A. The contribution of JED to ARCTAS was
supported by NASA grant NNX08AH69G.



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





Table 1. Lognormal mode parameters ($r_g$ in µm, $\sigma$ unitless), density ($\rho$, g cm$^{-3}$) and refractive
indices ($n$, unitless) at 550 nm and 0% RH of selected aerosol types from OPAC v3.1, GMI,
GEOS-Chem v9-02, and GC-RT.

| Model | Parameter | BC/Soot | OC/WASO | Sea Salt (Acc. Mode) | Trop. Sulfate |
|---|---|---|---|---|---|
| OPAC v3.1 | $r_g$ | 0.0118 | 0.0212 | 0.2090 | 0.0695 |
| | $\sigma$ | 2.00 | 2.24 | 2.03 | 2.03 |
| | $\rho$ | 1.0 | 1.8 | 2.2 | 1.7 |
| | $n$ | 1.75-0.44$i$ | 1.53-0.006$i$ | 1.50-10$^{-8}i$ | 1.43-10$^{-8}i$ |
| GMI | $r_g$ | 0.0118 | 0.0212 | 0.2090 | 0.05 |
| | $\sigma$ | 2.0 | 2.0 | 2.0 | 2.0 |
| | $\rho$ | 1.5 | 1.5 | 2.2 | 1.769 |
| | $n$ | 1.75-0.44$i$ | 1.53-0.006$i$ | 1.50-10$^{-8}i$ | 1.43-10$^{-8}i$ |
| GEOS-Chem v9-02 | $r_g$ | 0.02 | 0.063 | 0.09 | 0.07 |
| | $\sigma$ | 1.6 | 1.6 | 1.5 | 1.6 |
| | $\rho$ | 1.0 | 1.8 | 2.2 | 1.7 |
| | $n$ | 1.75-0.44$i$ | 1.53-0.006$i$ | 1.50-10$^{-8}i$ | 1.53-0.01$i$ |
| GC-RT[a] | $r_g$ | 0.02 | 0.063 | 0.09 | 0.07 |
| | $\sigma$ | 1.6 | 1.6 | 1.5 | 1.6 |
| | $\rho$ | 1.8 | 1.8 | 2.2 | 1.7 |
| | $n$ | 1.95-0.79$i$ | 1.53-0.006$i$ | 1.50-10$^{-8}i$ | 1.53-0.01$i$ |

[a]Baseline GC-RT configuration as described in Wang et al. (2014).



Table 2. Summary of results for modeling the submicron scattering coefficient throughout the
entire ARCTAS campaign with OPAC v3.1, GMI, GEOS-Chem v9-02, and GC-RT. The
mean ($\mu$) and standard deviation ($\sigma$) of the $\log_{10}$ error distributions are shown. The correlation
coefficient ($r^2$) and slope of the linear fit between the modeled and measured values are
shown as well.

| Wavelength | Metric | OPAC v3.1 | GMI | GEOS-Chem v9-02 | GC-RT[a] |
|---|---|---|---|---|---|
| | $\mu \pm \sigma$ | 0.07 ± 0.24 | 0.06 ± 0.24 | 0.15 ± 0.25 | 0.15 ± 0.24 |
| 450 nm | $r^2$ | 0.74 | 0.74 | 0.72 | 0.73 |
| | Slope | 0.89 ± 0.02 | 0.91 ± 0.02 | 1.01 ± 0.02 | 1.05 ± 0.02 |
| | $\mu \pm \sigma$ | 0.15 ± 0.24 | 0.09 ± 0.24 | 0.17 ± 0.25 | 0.16 ± 0.24 |
| 550 nm | $r^2$ | 0.72 | 0.72 | 0.72 | 0.73 |
| | Slope | 0.95 ± 0.02 | 0.94 ± 0.02 | 1.05 ± 0.02 | 1.08 ± 0.03 |
| | $\mu \pm \sigma$ | 0.27 ± 0.25 | -0.05 ± 0.27 | -0.04 ± 0.24 | -0.09 ± 0.25 |
| 700 nm | $r^2$ | 0.70 | 0.58 | 0.72 | 0.71 |
| | Slope | 1.19 ± 0.03 | 0.63 ± 0.01 | 0.68 ± 0.02 | 0.63 ± 0.02 |

[a]Baseline GC-RT configuration as described in Wang et al. (2014).





Table 3. Summary of results for modeling the submicron absorption coefficient (using the
correction of Lack et al., 2008) throughout the entire ARCTAS campaign with OPAC v3.1,
GMI, GEOS-Chem v9-02, and GC-RT. The mean ($\mu$) and standard deviation ($\sigma$) of the $\log_{10}$
error distributions are shown. The correlation coefficient ($r^2$) and slope of the linear fit
between the modeled and measured values are shown as well.

| Wavelength | Metric | OPAC v3.1 | GMI | GEOS-Chem v9-02 | GC-RT[a] |
|---|---|---|---|---|---|
| | $\mu \pm \sigma$ | $0.12 \pm 0.28$ | $0.04 \pm 0.27$ | $0.26 \pm 0.26$ | $0.20 \pm 0.26$ |
| 470 nm | $r^2$ | 0.70 | 0.72 | 0.69 | 0.70 |
| | Slope | $0.81 \pm 0.06$ | $0.83 \pm 0.04$ | $1.09 \pm 0.05$ | $0.96 \pm 0.04$ |
| | $\mu \pm \sigma$ | $0.28 \pm 0.29$ | $0.06 \pm 0.27$ | $0.27 \pm 0.25$ | $0.22 \pm 0.25$ |
| 532 nm | $r^2$ | 0.70 | 0.71 | 0.68 | 0.69 |
| | Slope | $0.94 \pm 0.06$ | $0.94 \pm 0.05$ | $1.21 \pm 0.06$ | $1.07 \pm 0.06$ |
| | $\mu \pm \sigma$ | $0.14 \pm 0.29$ | $0.00 \pm 0.24$ | $0.15 \pm 0.25$ | $0.09 \pm 0.24$ |
| 660 nm | $r^2$ | 0.68 | 0.68 | 0.65 | 0.67 |
| | Slope | $0.99 \pm 0.10$ | $0.75 \pm 0.04$ | $0.95 \pm 0.06$ | $0.84 \pm 0.05$ |

[a]Baseline GC-RT configuration as described in Wang et al. (2014).



Table 4. Summary of results for modeling the submicron single scattering albedo (SSA, using
the correction of Lack et al., 2008) throughout the entire ARCTAS campaign with OPAC
v3.1, GMI, GEOS-Chem v9-02, and GC-RT. The mean ($\mu$) and standard deviation ($\sigma$) of the
absolute error distributions are shown. The correlation coefficient ($r^2$) and slope of the linear
fit between the modeled and measured values are shown as well.

| Wavelength | Metric | OPAC v3.1 | GMI | GEOS-Chem v9-02 | GC-RT[a] |
|---|---|---|---|---|---|
| 450 nm | $\mu \pm \sigma$ | -0.01 ± 0.05 | 0.01 ± 0.05 | -0.01 ± 0.05 | 0.00 ± 0.05 |
| | $r^2$ | 0.17 | 0.07 | 0.09 | 0.10 |
| | Slope | 0.44 ± 0.03 | 0.65 ± 0.05 | 0.70 ± 0.04 | 0.61 ± 0.04 |
| 550 nm | $\mu \pm \sigma$ | -0.01 ± 0.05 | 0.01 ± 0.05 | -0.01 ± 0.05 | -0.01 ± 0.05 |
| | $r^2$ | 0.15 | 0.06 | 0.10 | 0.10 |
| | Slope | 0.43 ± 0.04 | 0.69 ± 0.06 | 0.73 ± 0.04 | 0.66 ± 0.04 |
| 700 nm | $\mu \pm \sigma$ | 0.01 ± 0.06 | 0.02 ± 0.07 | -0.02 ± 0.06 | -0.02 ± 0.06 |
| | $r^2$ | 0.14 | 0.03 | 0.11 | 0.10 |
| | Slope | 0.32 ± 0.02 | 0.78 ± 0.14 | 0.79 ± 0.04 | 0.81 ± 0.04 |

[a]Baseline GC-RT configuration as described in Wang et al. (2014).




Table 5.Summary of results for modeling the submicron scattering coefficient throughout the
entire ARCTAS campaign for ASP v2.1 using different mixing states. The mean ($\mu$) and
standard deviation ($\sigma$) of the $\log_{10}$ error distributions are shown. The correlation coefficient
($r^2$) and slope of the linear fit between the modeled and measured values are shown as well.

| Wavelength | Metric | ASP v2.1 CS | ASP v2.1 EXT | ASP v2.1 VA | ASP v2.1 MG |
|---|---|---|---|---|---|
| | $\mu \pm \sigma$ | -0.03 ± 0.17 | -0.02 ± 0.17 | -0.03 ± 0.17 | -0.03 ± 0.17 |
| 450 nm | $r^2$ | 0.89 | 0.89 | 0.89 | 0.89 |
| | Slope | 0.99 ± 0.02 | 1.01 ± 0.02 | 0.99 ± 0.02 | 1.00 ± 0.02 |
| | $\mu \pm \sigma$ | -0.03 ± 0.17 | -0.02 ± 0.17 | -0.03 ± 0.17 | -0.03 ± 0.17 |
| 550 nm | $r^2$ | 0.89 | 0.89 | 0.89 | 0.89 |
| | Slope | 1.00 ± 0.02 | 1.02 ± 0.02 | 1.00 ± 0.02 | 1.01 ± 0.02 |
| | $\mu \pm \sigma$ | -0.01 ± 0.17 | 0.00 ± 0.17 | -0.01 ± 0.17 | 0.00 ± 0.17 |
| 700 nm | $r^2$ | 0.88 | 0.88 | 0.88 | 0.88 |
| | Slope | 1.07 ± 0.03 | 1.10 ± 0.03 | 1.08 ± 0.03 | 1.08 ± 0.03 |






Table 6. Summary of results for modeling the submicron absorption coefficient (using the
correction of Lack et al., 2008) throughout the entire ARCTAS campaign for ASP v2.1 using
different mixing states. The mean ($\mu$) and standard deviation ($\sigma$) of the $\log_{10}$ error
distributions are shown. The correlation coefficient ($r^2$) and slope of the linear fit between the
modeled and measured values are shown as well.

| Wavelength | Metric | ASP v2.1 CS | ASP v2.1 EXT | ASP v2.1 VA | ASP v2.1 MG | ASP v2.1 Variable |
|---|---|---|---|---|---|---|
| | $\mu \pm \sigma$ | $0.11 \pm 0.32$ | $-0.10 \pm 0.32$ | $0.16 \pm 0.30$ | $0.07 \pm 0.32$ | $0.04 \pm 0.28$ |
| 470 nm | $r^2$ | 0.47 | 0.50 | 0.47 | 0.47 | 0.45 |
| | Slope | $0.85 \pm 0.09$ | $0.54 \pm 0.06$ | $0.90 \pm 0.09$ | $0.79 \pm 0.08$ | $0.60 \pm 0.08$ |
| | $\mu \pm \sigma$ | $0.13 \pm 0.32$ | $-0.05 \pm 0.31$ | $0.17 \pm 0.30$ | $0.09 \pm 0.32$ | $0.07 \pm 0.28$ |
| 532 nm | $r^2$ | 0.46 | 0.48 | 0.47 | 0.47 | 0.44 |
| | Slope | $0.93 \pm 0.12$ | $0.62 \pm 0.09$ | $0.98 \pm 0.13$ | $0.85 \pm 0.11$ | $0.68 \pm 0.11$ |
| | $\mu \pm \sigma$ | $0.13 \pm 0.32$ | $0.00 \pm 0.32$ | $0.17 \pm 0.30$ | $0.09 \pm 0.32$ | $0.09 \pm 0.29$ |
| 660 nm | $r^2$ | 0.46 | 0.47 | 0.46 | 0.46 | 0.44 |
| | Slope | $0.97 \pm 0.16$ | $0.72 \pm 0.13$ | $1.02 \pm 0.16$ | $0.89 \pm 0.14$ | $0.76 \pm 0.14$ |




Table 7. Summary of results for modeling the SSA (using the correction of Lack et al., 2008)
throughout the entire ARCTAS campaign for ASP v2.1 using different mixing states. The
mean ($\mu$) and standard deviation ($\sigma$) of the absolute error distributions are shown. The
correlation coefficient ($r^2$) and slope of the linear fit between the modeled and measured
values are shown as well.

| Wavelength | Metric | ASP v2.1 CS | ASP v2.1 EXT | ASP v2.1 VA | ASP v2.1 MG | ASP v2.1 Variable |
|---|---|---|---|---|---|---|
|  | $\mu \pm \sigma$ | -0.02 ± 0.04 | 0.01 ± 0.03 | -0.03 ± 0.04 | -0.01 ± 0.04 | -0.01 ± 0.04 |
| 450 nm | $r^2$ | 0.20 | 0.18 | 0.24 | 0.20 | 0.30 |
|  | Slope | 0.51 ± 0.06 | 0.30 ± 0.03 | 0.64 ± 0.07 | 0.46 ± 0.05 | 0.61 ± 0.06 |
|  | $\mu \pm \sigma$ | -0.04 ± 0.04 | -0.01 ± 0.03 | -0.04 ± 0.04 | -0.03 ± 0.04 | -0.03 ± 0.04 |
| 550 nm | $r^2$ | 0.20 | 0.17 | 0.25 | 0.20 | 0.30 |
|  | Slope | 0.54 ± 0.06 | 0.32 ± 0.03 | 0.67 ± 0.06 | 0.48 ± 0.05 | 0.64 ± 0.06 |
|  | $\mu \pm \sigma$ | -0.02 ± 0.05 | 0.00 ± 0.05 | -0.03 ± 0.05 | -0.01 ± 0.05 | -0.01 ± 0.05 |
| 700 nm | $r^2$ | 0.17 | 0.13 | 0.22 | 0.16 | 0.25 |
|  | Slope | 0.44 ± 0.05 | 0.28 ± 0.03 | 0.55 ± 0.05 | 0.38 ± 0.04 | 0.52 ± 0.05 |





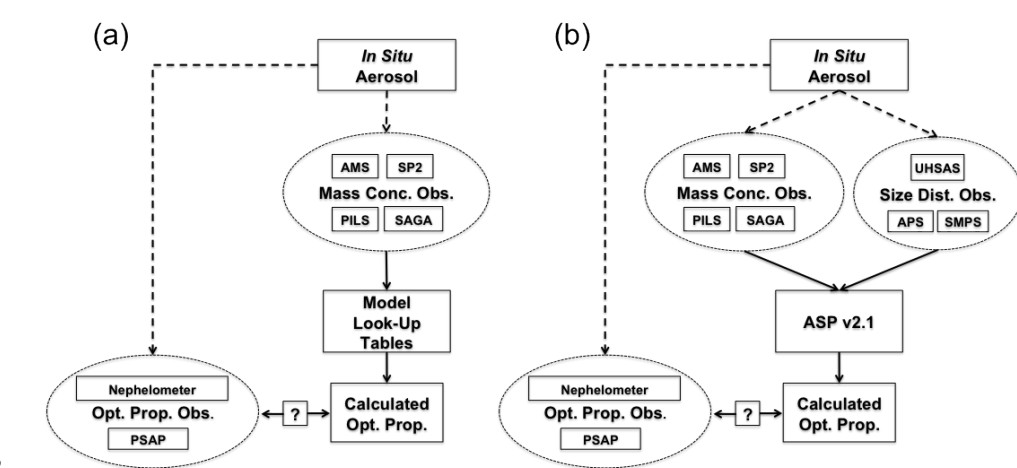

4  Figure 1. (a) Schematic of aerosol optical property closure study methodology for the OPAC

5  v3.1, NASA GMI Combo model, GEOS-Chem v9-02, and baseline GC-RT aerosol

6  parameterizations. (b) Schematic of closure study for ASP v2.1.





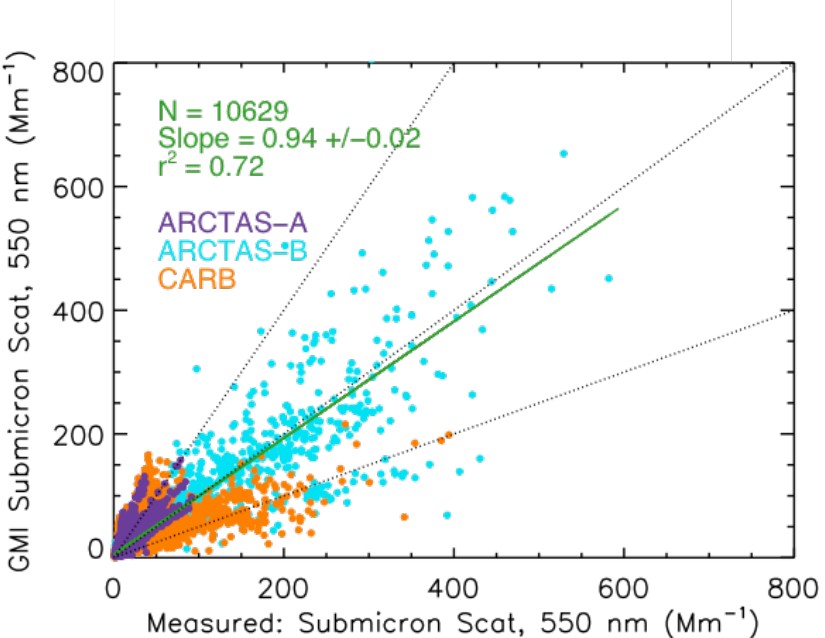

Figure 2. Example scatterplot of the measured submicron scattering coefficient (Mm$^{-1}$) at 550
4  nm versus the calculated submicron scattering coefficient for the GMI model. The color of the
data points corresponds to the phase of the ARCTAS campaign (ARCTAS-A in purple,
ARCTAS-B in cyan, and ARCTAS–CARB in orange). The dotted black lines are the 1:1 line,
2:1 line, and 1:2 line. The green line is the linear fit to the data. The number of data points
($N$), the slope of the linear fit, and the correlation coefficient ($r^2$) are shown as well.





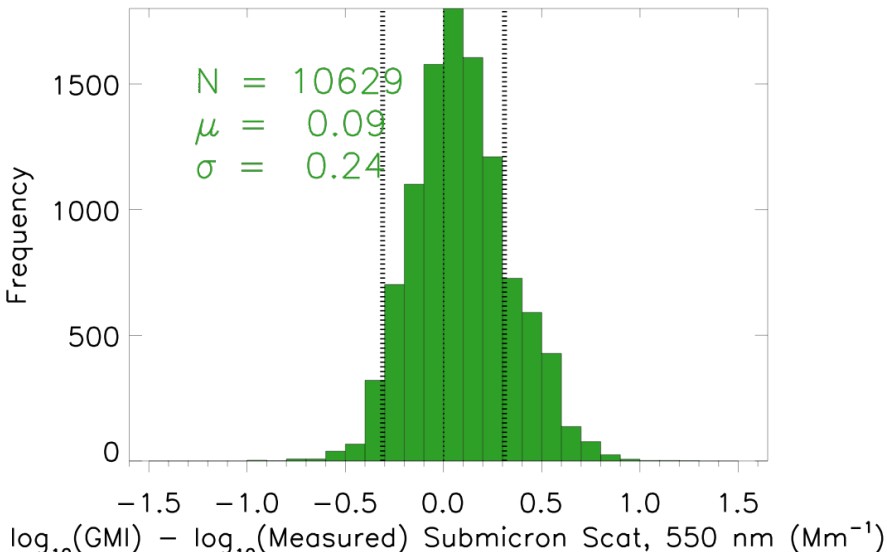

Figure 3. Example histogram of the errors in the $\log_{10}$ of the submicron scattering coefficient
(modeled value – measured value) at 550 nm for the GMI model. The vertical dashed lines
are at ± 0.31, corresponding to an error of a factor of 2. The number of data points ($N$) and the
mean ($\mu$) and standard deviation ($\sigma$) of the error distribution are also shown.





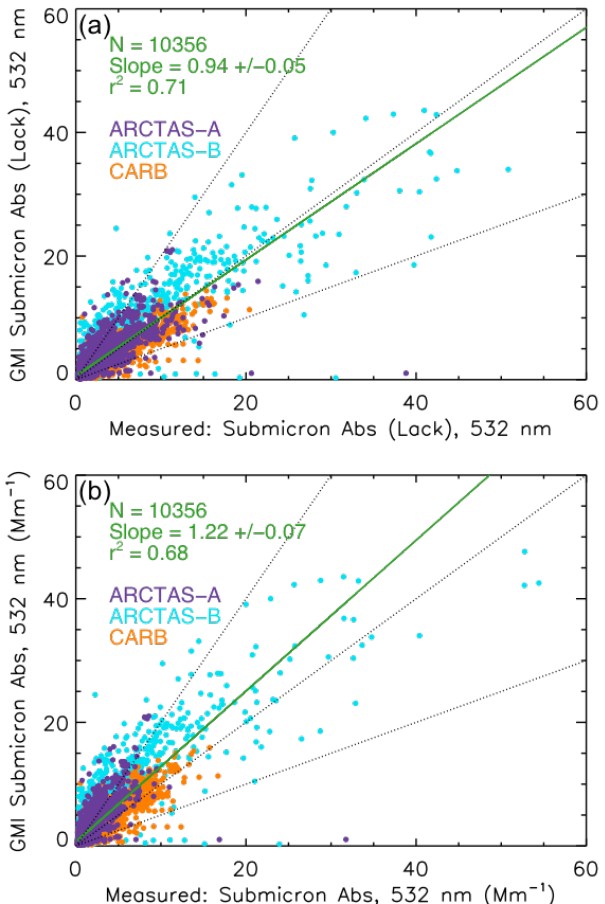

3    Figure 4. As in Figure 2, but for submicron aerosol absorption coefficients (Mm⁻¹) at 532 nm.

4    The PSAP measured aerosol absorption has been corrected following (a) Lack et al. (2008)

5    and (b) Virkkula (2010).





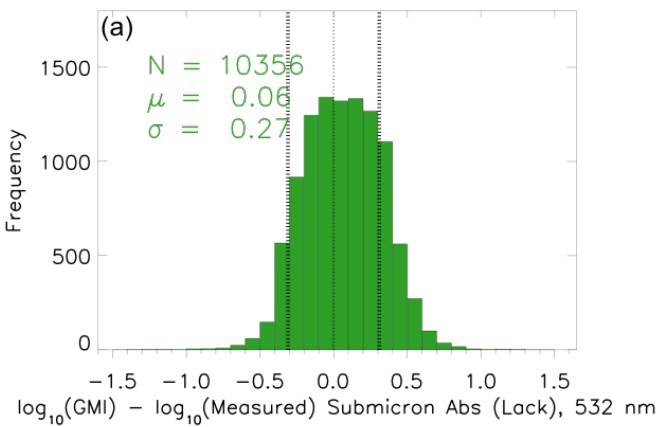

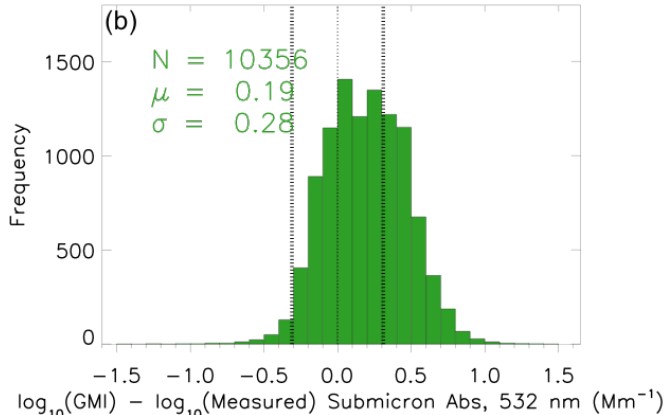

3    Figure 5. As in Figure 3, but for submicron aerosol absorption coefficients (Mm$^{-1}$) at 532 nm.

4    The PSAP measured aerosol absorption has been corrected following (a) Lack et al. (2008)

5    and (b) Virkkula (2010).





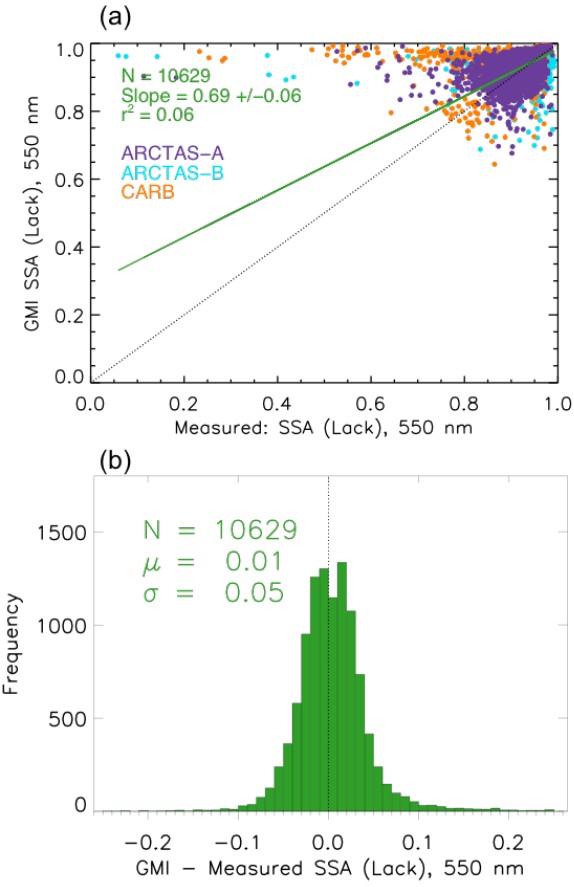

Figure 6. (a) Scatterplot of the measured submicron single scattering albedo (SSA) at 550 nm
versus the calculated submicron SSA for the GMI model. The dotted black line is the 1:1 line.
The green line is the linear fit to the data. (b) Histogram of the errors in the SSA (modeled
value – measured value) at 550 nm for the GMI model.



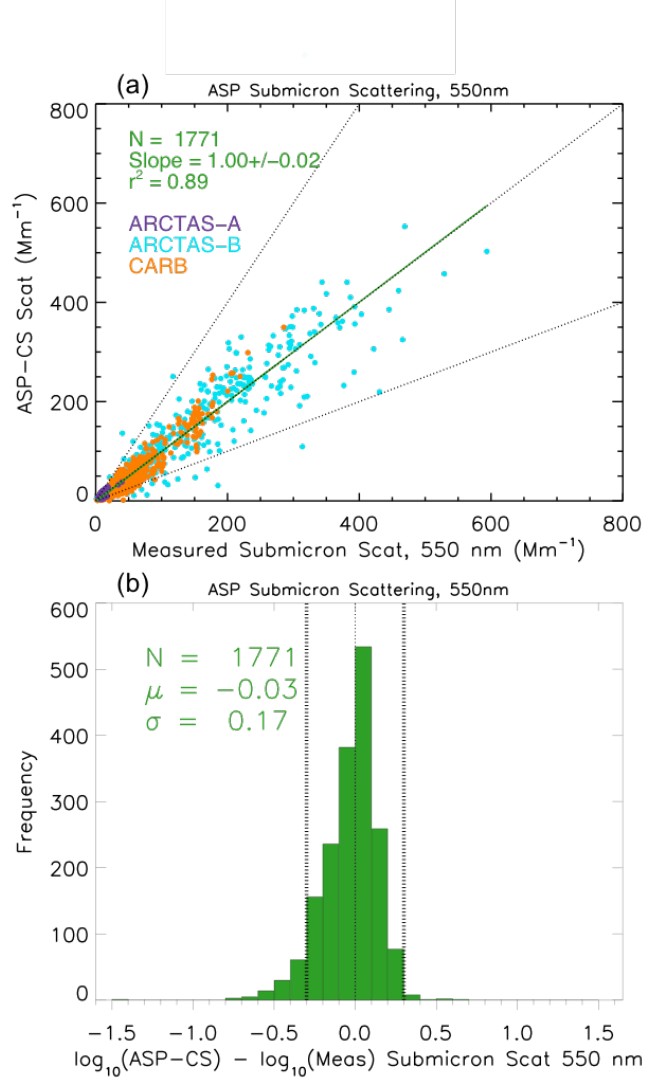

Figure 7. (a) Scatterplot of the measured submicron scattering coefficient (Mm$^{-1}$) at 550 nm
versus the calculated submicron scattering coefficient for ASP v2.1. The dotted black lines
are the 1:1, 2:1, and 1:2 lines. The green line is the linear fit to the data. Results for core-shell
(CS) mixing state are shown, but the results for other mixing states are similar (see Table 5).
(b) Histogram of the errors in the log$_{10}$ of the submicron scattering coefficient (modeled value
– measured value) at 550 nm for ASP v2.1. The vertical dashed lines are at ± 0.31,
corresponding to an error of a factor of 2.





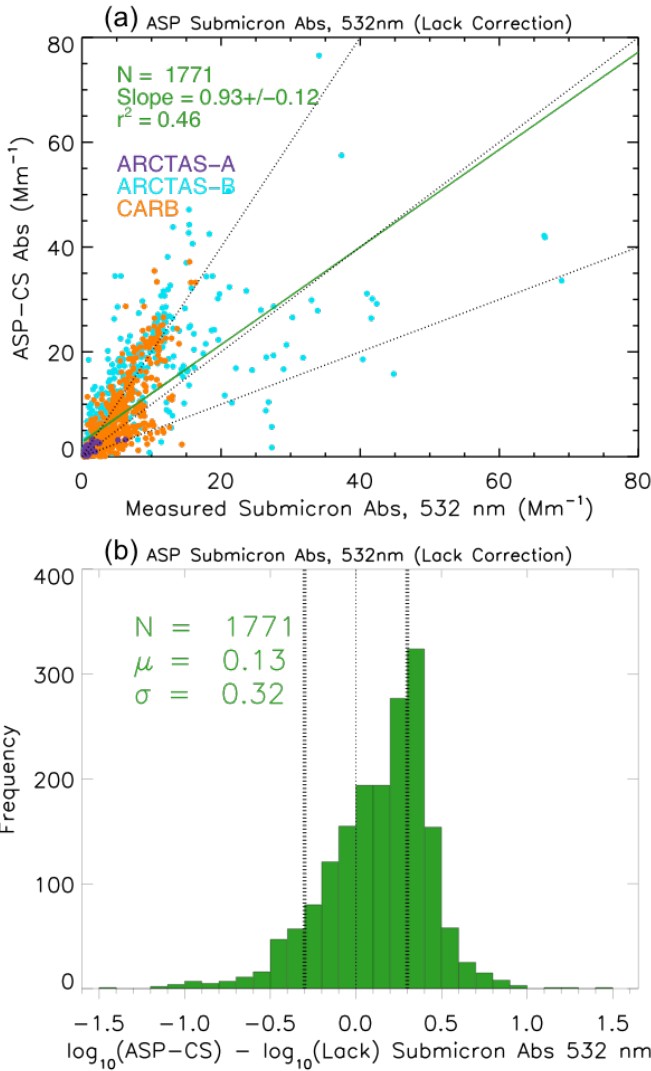

Figure 8. (a) Scatterplot of the measured submicron absorption coefficient (Mm$^{-1}$) at 532 nm

versus the calculated submicron absorption coefficient for ASP v2.1. These results use the

core-shell (CS) mixing state. (b) Histogram of the errors in the log$_{10}$ of the submicron

absorption coefficient (modeled value – measured value) at 550 nm for ASP v2.1 with the CS

mixing state.



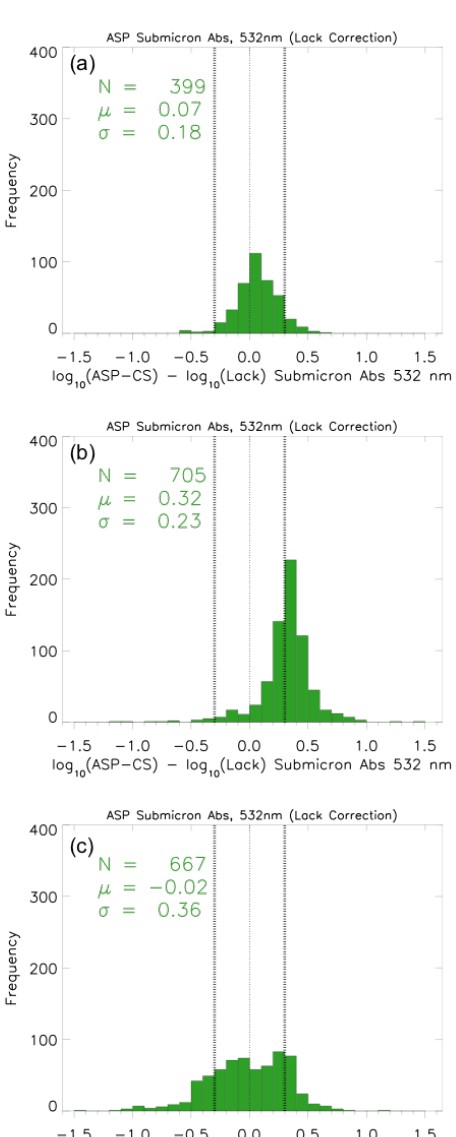

Figure 9. Histogram of the errors in the $\log_{10}$ of the submicron absorption coefficient
(modeled value – measured value) at 550 nm for ASP v2.1 with the CS mixing state. The
PSAP measured aerosol absorption has been corrected following Lack et al. (2008). Results
are broken apart for (a) ARCTAS-A, (b) ARCTAS-B, and (c) ARCTAS-CARB.





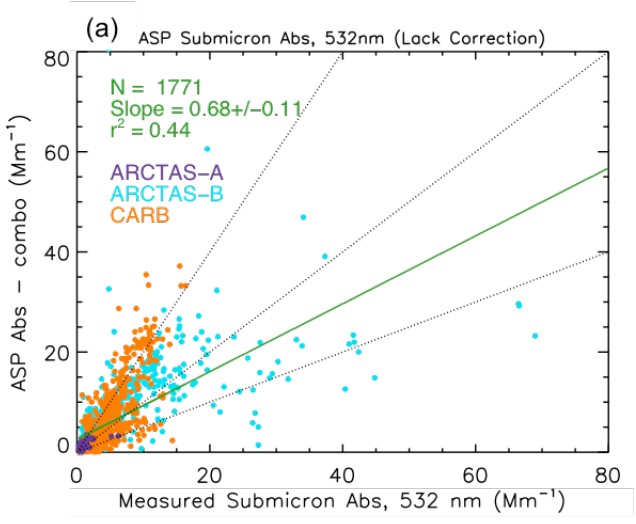

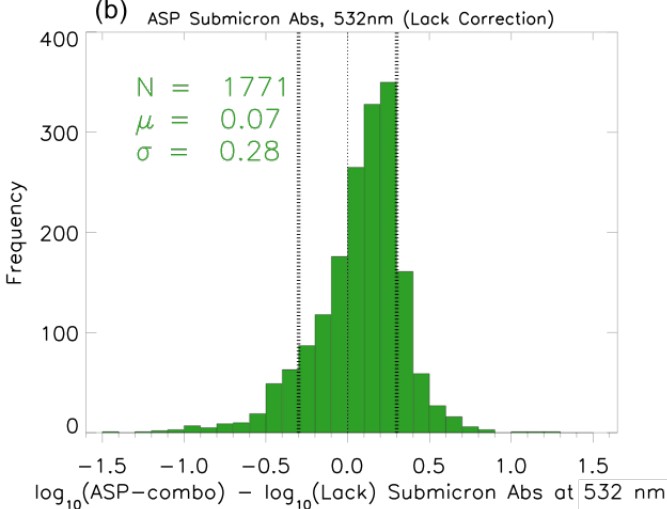

3    Figure 10. (a) As in Figure 8a, but using a variable mixing state - core-shell (CS) is used for

4    ARCTAS-A and ARCTAS-CARB, while an external mixture (EXT) is used for ARCTAS-B.

5    (b) As in Figure 8b but with this variable mixing state.

