# Peer review of "Evaluating Model Parameterizations of Submicron Aerosol"

_Atmospheric Chemistry and Physics, 2015_

## Referee Comment (RC1) · Anonymous Referee #1 · 27 Feb 2016

This paper describes a closure study that examines the performance of aerosol optical property treatments represented by several models when driven by observed aircraft measurements. Given that climate models have uncertainties in simulating aerosol radiative forcing and chemical transport models are sensitive to radiation via photoloysis rates, this study is important in highlighting deficiencies in models that could be remedied in part by adopting more state-of-the-science treatments of aerosols. The methodology employed is similar to previous closure studies, but as far as I know the models in question have not been confronted with the type of measurements before. Despite the importance of the results of such a closure study, I have several concerns regarding specific details of the approach that affect their conclusions.

[Figure]

Major Comments:

1) A number of times in the manuscript the authors note that computational expense of optical properties as being a reason for simpler approaches employed. It would be useful to be more specific and actually quantify the differences in computational expense of the treatments in the four simpler models versus the ASP model. I have used Mie calculations in 3-D models for many years, and I have never thought of it being a large computational burden compared to other components of a chemistry model. My impression that one reason for many models not adopting new treatments for optical properties is that the scientific community has focused more effort on "hot topics" (e.g. SOA) that overshadow old topics that are viewed as closed and get relatively little attention. In addition there are "simplified" Mie calculations described in the literature.

2) In section there is about a paragraph of information to describe the optical properties used by the four global models used in this study, but I did not get enough information to understand how they compute aerosol optical properties as well as understanding the differences between models. The material seems out of balance with the material presented for ASP. There are two pages of information describing the entire ASP model that has little to do with aerosol optical properties and is really not needed later in the manuscript. I assume that much of the background material on ASP has been described elsewhere and could be greatly shortened. I suggest the entire section be revised to better communicate the methodology of the optical properties for all five models.

3) There is a lot of scatter associated with closure study for absorption that may have nothing to do with problems in the optical property treatments used in the models. I strongly suggest the authors compare the time series of absorption and black carbon and determine how well those two quantities correlate, and include some subset of this analysis in the paper. One possible issue is that the PSAP may not work well for low concentrations of BC, and if that is the case those data points should not be used in the closure analyses.

4) It would have been useful to propagate some of the measurement and optical property treatment uncertainties into the closure study. Considering the consequences of the measurement uncertainties, and some are no small, are particularly important since the point of the study is to assess how well the model treatments perform.

Specific comments:

Page 2, lines 21-22: "Adding in situ size distribution ..." sounds awkward to me and should be rephrased. I think what the authors are trying to say is that ASP accounts for aerosol size distribution, but the other models treat aerosols as bulk species with no size distribution? Or is the difference fixed versus variable size distributions?

Page 2, lines 24-25: "mixing state" is used here when I think the authors mean mixing rule (as said earlier on line 16). Throughout the text the authors then seem to suggest that these mixing rules are mixing states – and the authors should use language that does not confuse readers. To me whether a model treats complex mixing states has to do more with just the optical properties – it is a fundamental part of how aerosol thermodynamics and dynamics are handled. Simple aerosol models that treat bulk aerosol (either in the simplest external or internal states) can manipulate model parameters using mixing rules (e.g. changing from external state to shell-core) even when that model is not simulating aerosol properties that way. In addition, there are only a limited number of mixing rules that do not reflect the continuum of mixing states.

Page 3, line 30: The authors state that the direct effect is uncertain which is true, but the uncertainties in the indirect effect are far larger and are of more concern the climate modelers.

Page 4, line 9: Change "intensity" to "expense".

Page 4, lines 29-31: Those conclusions are drawn using data collected during dry conditions and my not apply under higher RH.

Page 5, lines 14-16: I assume that a combination of these factors also contribute to the

errors?

Page 11, line 18: The authors mention the UHSAS instrument, which is an optical counter measurement that depend on an assumed refractive index (non-absorbing) that may not be representative of the ambient conditions being measured (see Kassianov et al. Atmosphere, 2015). But the authors do not describe whether the ARCTAS data has been corrected for this artifact. The authors should determine if the data has been corrected and describe some of the uncertainties in the measurements that will affect their closure study.

Page 12, lines 1-2: A reference is needed for these measurement uncertainties. Measurement uncertainties quoted in the literature for the AMS instrument varies widely, unfortunately.

Page 12, line 28: The statement regarding the aerosol mass to be dominated by OA and BC is strange. BC is usually very small, except in the center of biomass plumes. I would still expect sulfate to be a large fraction of mass. In fact, the AMS measurements should tell you whether there is significant sulfate (or nitrate and ammonium too) or not.

Page 13: end of Section 3.2: I would like to see a figure showing the merged size distribution and the individual size distributions from the SMPS, UHSAS, and APS. The critical part is merging these distributions where they overlap. I suspect that at individual times, the overlap might have large disagreements – but that when the data is averaged over a period of time those disagreements likely become smaller.

Page 14: Section 3.3: More discussion is needed regarding the uncertainties in the scattering and absorption measurements. For PSAP, there have been a number of studies pointing out some of the flaws with the instrument.

Page 17, lines 1-8: More description is needed on how the aerosol composition is used. I understand the size distributed AMS data is either not available or it may not be desirable to average the optical property measurements over an hour period. But

the size distribution for organic matter, sulfate, nitrate, and ammonium may not be the same. If they are assumed to be the same this is an important piece in the methodology to mention.

Page 18: lines 23-30: I do not see where data for the 450 nm channel are presented.

Page 20, lines 3-16: There are studies that suggest that uncertainties in SSA needs to be very small to have confidence in radiative forcing estimates. The results from this study show that even with the observations we cannot achieve this (large amounts of scatter). So what does this mean for climate model calculations that will have even larger errors in size, composition, and mass? Some perspective on SSA accuracy needed for climate models based on from other studies needs to be discussed.

Page 22, line 7: The authors suggest that BC is expected to be externally mixed for fresh smoke. I suggest the authors actually provide evidence for this. A number of studies have shown that SP2 can be analyzed to produce coating thickness as a function of size. If the authors are correct then the coating thickness should be small to zero when the aircraft is sampling in fresh smoke, and larger thickness further downwind. Is coating thickness available from the ARCTAS SP2 measurements

Page 22, line 13: I do not think enough proof is provided regarding mixing state in the observations.

Page 22, line 26: Table 7 is presenting the results by mixing rules, not mixing state.

Page 24, line 12 and 20: I do not think the model does a reasonable job in absorption, contrary to what the authors state in line 12. Then in line 20 the authors state that there should be improvements in absorption – a contradiction in this paragraph.

---

## Referee Comment (RC2) · Anonymous Referee #3 · 15 Apr 2016

This paper presents a closure study using in situ observations of aerosol composition and size as well as aerosol optical properties during the ARCTAS campaign to validate five aerosol optical properties modules. During this campaign there was a large abundance of biomass burning aerosol, so the focus of this closure study is on absorption. Four of those modules have in common that they prescribe the aerosol size distribution parameters, so the aerosol mass concentrations are the only inputs. These modules are used in current global models. The fifth optical properties module is used as part of the more detailed aerosol process model ASP and can make use of size distribution information.

I regard this paper as a valuable contribution to isolate and quantify errors in optical

properties calculations that are part of state-of-the art models. I have some questions and suggestions that would make the presentation of this paper clearer. I recommend this paper for publication after these are addressed.

**General comments**

1. Closure studies of optical properties have a long history. I encourage the authors to shift the emphasis in the introduction from describing the importance of LAC to reviewing some of the closure studies in the literature to provide context for the work under review. A few examples are Cai et al., 2011, Highwood et al., 2012, Esteve et al., 2014, Quinn and Coffman, 1998, Sciare et al., 2005, Wex et al., 2002, but there are many more.

Cai et al., Journal of Geophysical Research, 116 (2011), D02202

Highwood et al., Atmospheric Chemistry and Physics, 12 (2012), 7251–7267

Esteve et al., Atmospheric Environment, 89 (2014), 517- 524

Quinn and Coffman, Journal of Geophysical Research, 103 (1998), 16575–16596

Sciare et al., Atmospheric Chemistry and Physics, 5 (2005), 2253–2265

Wex et al., Journal of Geophysical Research, 107 (2002), 8122

2. The description of the models/ aerosol optical property modules is somewhat confusing. I recommend referring consistently to the actual optical property module in each case, since the full chemical transport models are not used for this study. For example, in section 2.2, what is actually evaluated is a portion of the FAST-JX model, correct? Similarly, large parts of section 2.5 seem to be not relevant for this paper, as there are no model runs using the full ASP model, but only the optical properties module is used. If that's indeed the case, I recommend eliminating or substantially shortening section 2.5.1, except for the description about the assumptions how aerosol composition is represented over the different size bins.

**Specific comments**

1. Replace the phrase "to get" with "to obtain" (e.g. line 5)

2. page 4, line 9: replace "intensity" with "burden" or similar

3. page 5, line 2: should read "result"

4. page 6, line 7: suggest to rephrase: "six types of water clouds, three types of ice clouds and ten aerosol types"

5. page 6, line 12: The list of different components is confusing. Is water-insoluble aerosol everything insoluble except for soot and mineral dust (which are a separate class), and is water-soluble aerosol everything soluble except for sea salt and sulfate? Please clarify.

6. page 6, line 30: should read "based on Mie theory calculations" (remove "the").

7. page 7, line 1: How is dependence of refractive indices on relative humidity parameterized?

8. page 8, line 24: suggest to rephrase: "all particles in a size bin are assumed to have the same composition".

9. Page 8, line 26: suggest to rephrase: "one bin each for particles smaller than 10 nm or larger than 20 $\mu$m."

10. page 8, line 28: This sentence says that the mass fractions of different aerosol components are the same for all the bins. Does this mean that the aerosol is assumed to be fully internally mixed? In other words, it's not only assumed that the particles within one bin have the same composition as stated two lines above, but all particles have the same composition?

11. page 10, line 27: Regarding the Maxwell Garnett mixing rule, are there assumptions necessary how many BC inclusions there are, if so, what is assumed in this

study?

12. page 11, line 22: The SP2 instrument only covers a range above 90 nm, but a fraction of BC particles will have smaller cores, which will be missing from the mass concentration, but their absorption will be captured. Can you estimate how big of a problem this is for the closure?

13. page 13, line 13: replace slower with lower.

14. page 13, line 19: should read 7.75 ?

15. Section 4.1: Since size distribution measurements are available, can you include some information of how close/how different the observed size distributions were from the assumed distributions?

16. page 15, line 16: add "volume" to "extinction coefficient"

17. page 16, line 26: here the authors talk about modes as inputs for ASP, which is in contradiction to the information on page 8, which talked about size bins. Please clarify.

18. page 19, line 2: delete "the shows"

19. Section 6: the term mixing state is used when it should be mixing rule.

20. Conclusions: The four fixed-size-distribution models are all rather similar. Can you comment on why the GMI leads to the best results?

21. page 24, line 13: What improvements might this be?

---

## Author Comment (AC1) · 16 Jun 2016

We thank the anonymous reviewers for their helpful comments on our paper. As detailed below, we have revised our manuscript to reflect their comments, which we feel has resulted in a stronger paper.

Below the reviewer's comments are in boldface, with our responses in plain text. Revised manuscript text is given in italics.

**Anonymous Referee #1**

**This paper describes a closure study that examines the performance of aerosol optical property treatments represented by several models when driven by observed aircraft measurements. Given that climate models have uncertainties in simulating aerosol radiative forcing and chemical transport models are sensitive to radiation via photoloysis rates, this study is important in highlighting deficiencies in models that could be remedied in part by adopting more state-of-the-science treatments of aerosols. The methodology employed is similar to previous closure studies, but as far as I know the models in question have not been confronted with the type of measurements before. Despite the importance of the results of such a closure study, I have several concerns regarding specific details of the approach that affect their conclusions.**

**Major Comments:**

**1) A number of times in the manuscript the authors note that computational expense of optical properties as being a reason for simpler approaches employed. It would be useful to be more specific and actually quantify the differences in computational expense of the treatments in the four simpler models versus the ASP model. I have used Mie calculations in 3-D models for many years, and I have never thought of it being a large computational burden compared to other components of a chemistry model. My impression that one reason for many models not adopting new treatments for optical properties is that the scientific community has focused more effort on "hot topics" (e.g. SOA) that overshadow old topics that are viewed as closed and get relatively little attention. In addition there are "simplified" Mie calculations described in the literature.**

We agree with the reviewer that on-line Mie calculations are not strictly prohibitive for 3D chemical transport models (CTMs). Rather, as the reviewer notes, given the desires for other model improvements that require additional computations (e.g., increased horizontal and vertical resolution, more chemical species and reactions, etc.), aerosol optical property calculations generally do not get priority for improvements that would require increased computational time. One of the goals of our study was to see if we could justify adding more computationally intensive approaches to 3D CTMs by demonstrating these improvements in ASP. Our results suggest that adding an on-line Mie calculation could help substantially for scattering calculations, but it is less clear that absorption calculations would be improved.

However, we do not think a direct timing test between the simpler models and ASP would be informative of the actual computational differences between the simpler approaches and an on-line Mie calculation, as ASP v2.1 calculates many more wavelengths (every 1 nm from 250 nm to 700 nm) than would be necessary for an on-line Mie calculation. Instead, we have revised our discussion of the relative computational burdens to make clear that the issue is not the absolute time required, but rather the decisions of many modeling groups to invest computational cycles in other processes. See P4, L12-13:

> *However, in order to reduce computational expense (so that the saved computational cycles can be used to increase model resolution, number of chemical species, etc.), most global chemical transport models (CTMs)…*

**2) In section there is about a paragraph of information to describe the optical properties used by the four global models used in this study, but I did not get enough information to understand how they compute aerosol optical properties as well as understanding the differences between models. The material seems out of balance with the material presented for ASP. There are two pages of information describing the entire ASP model that has little to do with aerosol optical properties and is really not needed later in the manuscript. I assume that much of the background material on ASP has been described elsewhere and could be greatly shortened. I suggest the entire section be revised to better communicate the methodology of the optical properties for all five models.**

Some of the background material on ASP is only available from the thesis of Alvarado (2008) rather than in the peer-reviewed literature, but we agree with both reviewers that it is out of place here and likely unnecessary as the thesis is available online. We have thus cut most of the discussion of the inorganic thermodynamics routines of ASP from the paper.

The optical property calculations of the global models are described in Sections 2.1 to 2.4 of the paper. The methods of OPAC v3.1 and GMI are described in detail, whereas the methods of GEOS-Chem v9-02 and GC-RT are similar to GMI except for the parameter changes noted in Table 1. We have added some text clarifying the methods:

P7, L9-11:

> *For the aerosol components analyzed in this paper (see Table 1), the optical properties are calculated from the assumed log-normal size distributions and refractive indices using Mie theory.*

P8, L10-12:

> *As in GMI, the optical properties for GEOS-Chem v9-02 are based on Mie theory calculations, but with changes to the assumed size distribution, refractive indices, and densities as noted in Table 1.*

P8, L26-27:

> *As in GMI and GEOS-Chem v9-02, the optical properties in GC-RT are based on Mie theory calculations.*

**3) There is a lot of scatter associated with closure study for absorption that may have nothing to do with problems in the optical property treatments used in the models. I strongly suggest the authors compare the time series of absorption and black carbon and determine how well those two quantities correlate, and include some subset of this analysis in the paper. One possible issue is that the PSAP may not work well for low concentrations of BC, and if that is the case those data points should not be used in the closure analyses.**

We have done the requested comparison for all the data points used in evaluating the fixed size distribution models. The correlation ($r^2$) between measured submicron absorption at 532 nm and measured submicron BC concentrations is 0.64, indicating a strong correlation (see Figure R1a below). When both data sets are plotted on a log scale (Figure R1b), there is some evidence that the measured absorption is independent of the measured BC concentration for BC concentrations below about 0.03 $\mu$g m$^{-3}$ (-1.5 on the x-axis of Figure R1b) which occurs for 261 out of 10,356 data points. However, excluding these points from the analysis does not significantly change the model performance statistics: for example, the slope, $r^2$, and mean bias for the green channel (532 nm) from GMI stay identical, while the standard deviation of the logarithm of the errors only changes from 0.27 to 0.26 (see Figures R2 and R3 below, and compare with Figures 4a and 5a in the paper).

[Figure]

**Figure R1. Submicron aerosol absorption coefficients using the Lack et al. (2008) correction versus the submicron BC mass measured using the SP2. The top panel is the absolute units, while the bottom panel shows the $\log_{10}$ of both.**

[Figure]

**Figure R2. Lack-corrected submicron absorption from the PSAP compared with the GMI model at 532 nm after filtering out data points with a measured BC mass below 0.03 μg m[-3].**

[Figure]

**Figure R3. Difference in the logarithms of the GMI and measured submicron absorption at 532 nm after filtering out data points with a measured BC mass below 0.03 μg m[-3].**

As we have no clear proof that the PSAP measurements are in error at low BC concentrations, and excluding them does not change our results, we prefer to keep these points in our closure study. However, we have added this discussion of potential errors in the PSAP at low BC concentrations to the text (P14, L23-29):

> *The PSAP submicron absorption measurement is well correlated with the submicron BC mass (not shown), with an overall $r^2$ of 0.64, but the data is uncorrelated for low BC mass concentrations ($<0.03$ $\mu g$ $m^{-3}$), which occurs for 261 out of 10,356 data points in our absorption analysis. However, as we have no clear evidence that these PSAP data points are incorrect and excluding them does not appreciably change the conclusions of our study, we have included these data points in our analysis below.*

**4) It would have been useful to propagate some of the measurement and optical property treatment uncertainties into the closure study. Considering the consequences of the measurement uncertainties, and some are no small, are particularly important since the point of the study is to assess how well the model treatments perform.**

We agree that including a rough uncertainty propagation would be useful to put the model results in context. Here we thus try to estimate what the errors would be for a hypothetical perfect model.

For the fixed size distribution models, the uncertainties in the inputs are primarily the uncertainties in the mass concentrations from the AMS and SP2. Taking the sum of the squares in the relative standard errors in the mass concentrations of OA (19%), sulfate (17%), nitrate (17%), ammonium (17%), chloride (17%), and BC (30%) gives an overall uncertainty in the inputs of ±49%.

For scattering, the errors in the nephelometer are estimated to be ±1% (Anderson and Ogren, 1998) or ±0.5 Mm$^{-1}$, whichever is larger. Thus we expect these errors to generally be negligible compared to the errors in the masses except at very small mass loadings. Thus we would expect that a perfect fixed-size distribution model would have an uncertainty of 49% in reproducing the observations. In our evaluation statistics this would correspond to a geometric standard deviation ($\sigma$) of 0.17. However, as we are looking at ~10,000 data points, the expected error in the mean is a factor of 100 lower. Thus a perfect model would have a mean ($\mu$) with an absolute value less than 0.01 with a $\sigma$ of 0.17.

For absorption, the optical measurement uncertainty is at least 30% due to uncertainty in the correct conversion from the filter-based PSAP measurement to the ambient absorption. Here we use the same input uncertainty in the masses, even though the model prediction will be most sensitive to errors in the absorbing components BC and OC. Thus for absorption, a perfect model would have an uncertainty of 57% in reproducing the observations, corresponding to a geometric standard deviation ($\sigma$) of 0.20 with a mean ($\mu$) with an absolute value less than 0.01.

For the ASP v2.1 closure studies, the uncertainty in the input size distributions is most important for scattering, with additional uncertainties due to the relative fraction of the different mass components in the aerosol. If we estimate the size distribution using the UHSAS number uncertainty of 20%, then that uncertainty alone would give a geometric standard deviation ($\sigma$) of 0.08. Estimating the impact of the relative mass errors is more complicated, as the total mass is constrained here by the size distribution measurements. If we assume the main impact on the scattering uncertainty is the estimated BC mass fraction (30%) and that the uncertainty in the scattering measurement is again negligible, a perfect model would have an uncertainty of 36% in reproducing the scattering observations, corresponding to a $\sigma$ of 0.13 with a geometric mean ($\mu$) with an absolute value less than 0.01.

For absorption, including the PSAP uncertainty (30%) would mean that a perfect model would have a an uncertainty of 47% in reproducing the absorption observations, corresponding to a $\sigma$ of 0.17 with a geometric mean ($\mu$) with an absolute value less than 0.01.

Based on this rough analysis, none of the models evaluated in this work are perfect, but it is clear that even a perfect model would have a reasonably large value for $\sigma$. We have added this context to the paper in the following places:

P19, L19-27:

*In order to interpret these closure study results, it is useful to estimate the values of $\mu$ and $\sigma$ we would expect from a perfect model based on the uncertainty in the input mass concentrations and the uncertainty in the scattering measurement. Taking the sum of the squares in the relative standard errors in the mass concentrations of OA (19%), sulfate (17%), nitrate (17%), ammonium (17%), chloride (17%), and BC (30%), as well as the estimated nephelometer uncertainty (1%) gives an overall uncertainty estimate of ±49%, corresponding to a $\sigma$ of 0.17. Thus even a perfect model would have a fairly significant spread in it's histogram of errors. However, as we are using ~10,000 data points in our comparison, the expected error in $\mu$ is about a factor of 100 lower, and thus we would expect $|\mu| \ll 0.01$.*

P20, L15-18:

*The expected values of $\mu$ and $\sigma$ from a perfect model are $\ll 0.01$ and ~0.20, respectively, reflecting the input uncertainties discussed for scattering in Section 5.1 as well as the ~30% uncertainty in converting the filter-based PSAP measurement to the ambient absorption.*

P22, L5-8:

*...we estimate that a perfect model would have a $\sigma$ of 0.13 due to the uncertainty in the size distributions and the relative mass contributions.*

P22, L18-20:

> *We estimate that a perfect model would have a σ of 0.17 due to the uncertainty in the size distributions, the relative mass contributions, and in the absorption measurement.*

**Specific comments:**

**Page 2, lines 21-22: "Adding in situ size distribution . . ." sounds awkward to me and should be rephrased. I think what the authors are trying to say is that ASP accounts for aerosol size distribution, but the other models treat aerosols as bulk species with no size distribution? Or is the difference fixed versus variable size distributions?**

We were referring to the fact that the four simplified models use fixed size distributions, while ASP has a variable size distribution that in this closure study is matched to the in situ size distribution measurements before calculating aerosol optical properties. We have revised this part of the abstract as follows (P2, L21-24):

> *Adding a variable size distribution, as in ASP v2.1, improves model performance for scattering but not for absorption, likely due to the assumption in ASP v2.1 that BC is present at a constant mass fraction throughout the aerosol size distribution.*

**Page 2, lines 24-25: "mixing state" is used here when I think the authors mean mixing rule (as said earlier on line 16). Throughout the text the authors then seem to suggest that these mixing rules are mixing states – and the authors should use language that does not confuse readers. To me whether a model treats complex mixing states has to do more with just the optical properties – it is a fundamental part of how aerosol thermodynamics and dynamics are handled. Simple aerosol models that treat bulk aerosol (either in the simplest external or internal states) can manipulate model parameters using mixing rules (e.g. changing from external state to shell-core) even when that model is not simulating aerosol properties that way. In addition, there are only a limited number of mixing rules that do not reflect the continuum of mixing states.**

The reviewer is correct, strictly speaking we are evaluating four different possible mixing rules in ASP, rather than modeling the aerosol mixing state explicitly. However, when we say that further development of ASP should focus on time varying mixing states, here we are referring to the explicit modeling of mixing state. We have clarified this in the abstract (P2, L24-27) and elsewhere in the revised manuscript.

**Page 3, line 30: The authors state that the direct effect is uncertain which is true, but the uncertainties in the indirect effect are far larger and are of more concern the climate modelers.**

We agree the uncertainties in the indirect affect are larger, but the uncertainties in the direct effect are not yet negligible. We have added the following statement to the end of this paragraph to clarify this (P4, L6-9):

> *While these uncertainties in the aerosol direct effect are generally smaller than the uncertainties in the forcing due to aerosol-cloud interactions (the indirect effect), they are still a significant cause of differences between climate models.*

**Page 4, line 9: Change "intensity" to "expense".**

Done.

**Page 4, lines 29-31: Those conclusions are drawn using data collected during dry conditions and my not apply under higher RH.**

We thank the reviewer for noting this, and have added the following sentence to this paragraph (P5, L2-4):

> *However, these observations were made under dry conditions and thus the result may not apply at higher RH.*

**Page 5, lines 14-16: I assume that a combination of these factors also contribute to the errors?**

We agree that interactions between these factors may also contribute to these errors, and have revised the text accordingly (P5, L19-21)

> *...to help determine if the errors in the global model routines are primarily due to their fixed size distributions, assumptions about external mixtures, their assumptions about the refractive indices of LAC, or interactions between these assumptions.*

**Page 11, line 18: The authors mention the UHSAS instrument, which is an optical counter measurement that depend on an assumed refractive index (non-absorbing) that may not be representative of the ambient conditions being measured (see Kassianov et al. Atmosphere, 2015). But the authors do not describe whether the ARCTAS data has been corrected for this artifact. The authors should determine if the data has been corrected and describe some of the uncertainties in the measurements that will affect their closure study.**

We agree with the reviewer that the actual refractive index of the particles may be different from the polystyrene latex spheres used to calibrate the UHSAS. Kassianov et al. (2015) present a method for accounting for the effect of the refractive index differences by correcting the size distributions from the UHSAS using an average RI calculated based on the measured aerosol composition. This correction was not applied to

the ARCTAS archived data set, and thus the size of the particles may be underestimated in the UHSAS, thereby underestimating the scattering. While this error might impact our results, Figure C2 in Kassianov et al. (2015) suggests that this correction is relatively small for the submicron aerosol of interest in this study. Furthermore, since refractive indices are one of the parameters we are attempting to evaluate, we would need to pick one of the model's assumed refractive indices to perform the correction, which would give that model an advantage in the intercomparison. Thus we have decided instead to discuss this uncertainty and the potential impacts in our paper (P13, L12-17):

> *Note that, as the UHSAS is an optical instrument, it can give incorrect size information if the refractive index of the particles is far from that of the polystyrene latex spheres used for calibration. This artifact can lead to a small underestimate of the size of submicron particles when it is not taken into account, thereby leading to an underestimate of scattering (e.g., Kassianov et al., 2015). However, we expect that this effect on our study will be small relative to the stated precision of the UHSAS.*

**Page 12, lines 1-2: A reference is needed for these measurement uncertainties. Measurement uncertainties quoted in the literature for the AMS instrument varies widely, unfortunately.**

The quoted uncertainties are those given in the NASA ARCTAS archive data files. The full statement is:
> UNCERTAINTY: Accuracy estimate (2sdev): Inorganics 34%, Organics 38%, dominated by uncertainty in the particle collection efficiency due to particle bounce (Eb, see Huffman et al., Aerosol Sci Technol. 39, 1143-1163, 2005). Precision at low concentrations is the same as the detection limits reported below. Precision at higher concentrations is much better than the accuracy. The accuracy estimates are likely conservative, and may be reduced when a more complete uncertainty analysis of the AMS ARCTAS dataset is completed.

To address this comment, we have added the Huffman et al. (2005) reference, as well as a reference to the Cubison et al. (2011) paper using the ARCTAS AMS data and Bahreini et al. (2009) (P11, L30).

**Page 12, line 28: The statement regarding the aerosol mass to be dominated by OA and BC is strange. BC is usually very small, except in the center of biomass plumes. I would still expect sulfate to be a large fraction of mass. In fact, the AMS measurements should tell you whether there is significant sulfate (or nitrate and ammonium too) or not.**

We agree with the reviewer – although the biomass burning aerosol sampled in ARCTAS-B are generally dominated by OC and BC, this is not always true for the ARCTAS-A and ARCTAS-CARB campaigns, where sulfate can be a significant fraction of the aerosol mass. What we were trying to say is that we don't expect that minor errors in our estimates of the concentrations of refractory cations from our procedure for

matching the SAGA and AMS observations will make a substantial difference, as we expect that most of the aerosol mass is already accounted for by the SP2 and AMS measurements. We have revised the text to make this clearer (P12, L27-30):

> *However, as we expect the aerosol in the ARCTAS campaign to be dominated by the species measured by the AMS and the SP2, errors in our estimates of the submicron refractory cation mass should have little impact on our closure study results.*

**Page 13: end of Section 3.2: I would like to see a figure showing the merged size distribution and the individual size distributions from the SMPS, UHSAS, and APS. The critical part is merging these distributions where they overlap. I suspect that at individual times, the overlap might have large disagreements – but that when the data is averaged over a period of time those disagreements likely become smaller.**

The method we used to combine the size distributions is described in Section 4.2 of the paper. Figure R4 below shows an example surface area size distribution for a single time point. The surface area distribution is shown here and used to fit the modes for ASP as scattering and absorption are expected to be functions of the particle's cross-sectional area. The "combined distribution" in the dashed line is constructed as described in the text, with the UHSAS data used for diameters between 60 nm and 850 nm, scaled SMPS data used for smaller sizes (with the scaling based on a regression of the points where the SMPS and UHSAS overlap) and with the decay of number concentration for diameters above 850 nm determined by a power-law fit to the APS data assuming a conversion factor of 0.8. We used the UHSAS as our primary source of size distribution data as it is most sensitive to the optically active submicron particles of interest in this study. Our study is less sensitive to the SMPS data (as the particles below 60 nm scatter and absorb relatively little light) and the APS data (as we focus on the optical properties of submicron aerosol).

[Figure]

**Figure 4. Plot of the surface area distribution (μm² cm⁻³) from the SMPS (red circles), UHSAS (green triangles), and APS (blue triangles for a correction factor of 1.0, yellow diamonds for a correction factor of 0.7) with the combined size distribution plotted as a black line.**

**Page 14: Section 3.3: More discussion is needed regarding the uncertainties in the scattering and absorption measurements. For PSAP, there have been a number of studies pointing out some of the flaws with the instrument.**

As noted in the text, the precision of the nephelometer was 0.5 Mm⁻¹ and the precision of the PSAP is 0.2 Mm⁻¹. The paper of Anderson and Ogren (1998) suggests that in practice the actual reproducibility of scattering measurements with a TSI model 3563 nephelometer is ±1%, so we have modified the text to reflect this uncertainty (P14, L8-11)

> *These total scattering coefficients were then corrected for truncation errors using the procedure described by Anderson and Ogren (1998), who report a measurement reproducibility of ±1%. Thus the actual uncertainty in the scattering meaurements is ±1% or ±0.5 Mm⁻¹, whichever is larger.*

In addition, we agree that, as a filter-based absorption measurement, the PSAP has some additional flaws in estimating the in situ absorption of aerosols, and as we discuss, the difference between the two corrections used in the ARCTAS study can differ by 20-30%,

which is likely closer to a true estimate of the uncertainty in the absorption measurement. We have edited the text to reflect this (P14, L21-22):

> *Thus the practical uncertainty in the absorption measurement is estimated as at least 20-30%.*

**Page 17, lines 1-8: More description is needed on how the aerosol composition is used. I understand the size distributed AMS data is either not available or it may not be desirable to average the optical property measurements over an hour period. But the size distribution for organic matter, sulfate, nitrate, and ammonium may not be the same. If they are assumed to be the same this is an important piece in the methodology to mention.**

The size distribution of these compounds is assumed to be the same in the ASP simulation, as suggested above, and we agree with the reviewer that this should be made clear in the methodology. We have addressed this by adding the following sentence to the end of this paragraph (P17, L22-24).

> *This implicitly assumes that the size distributions of all aerosol components (BC, OC, and electrolytes) are the same, which may not have been true in the ambient atmosphere.*

**Page 18: lines 23-30: I do not see where data for the 450 nm channel are presented.**

These results are given in Table 2, but that was ambiguous in the text. We have added a reference to Table 2 closer to this section in the revised paper (P19, L9).

**Page 20, lines 3-16: There are studies that suggest that uncertainties in SSA needs to be very small to have confidence in radiative forcing estimates. The results from this study show that even with the observations we cannot achieve this (large amounts of scatter). So what does this mean for climate model calculations that will have even larger errors in size, composition, and mass? Some perspective on SSA accuracy needed for climate models based on from other studies needs to be discussed.**

We agree that our results suggest that the instrumentation used in the ARCTAS campaign is not sufficient to constrain the aerosol SSA to the accuracy needed for climate studies, mainly due to the uncertainty in aerosol absorption. This points to the need for further development of instruments that can measure the absorption properties of aerosols in situ.

We also agree that additional model errors beyond the optical property calculations examined here, from processes such as emissions, chemistry, deposition, etc. could significantly increase the errors in aerosol scattering and absorption calculations in climate models. However, in situ data are not the only data available to constrain the direct radiative effect of aerosols in global models – as noted in our Introduction, AERONET data as well as a variety of satellite instruments (e.g., MODIS) have been used to constrain the net radiative effect of aerosols in these models.

In addition, we agree that some context on the needed accuracy in SSA for climate models would be helpful in interpreting our results. We have thus added the following text to our revised paper (P21, L7-11):

*Note that the size of the uncertainties in SSA seen in Table 4 can have a significant impact on estimates of global aerosol DRF. For example, Loeb and Su (2010) found that a SSA perturbation of 0.03 over land and 0.06 over ocean could lead to errors in all-sky DRF of -0.73 to +1.11 W m$^{-2}$.*

**Page 22, line 7: The authors suggest that BC is expected to be externally mixed for fresh smoke. I suggest the authors actually provide evidence for this. A number of studies have shown that SP2 can be analyzed to produce coating thickness as a function of size. If the authors are correct then the coating thickness should be small to zero when the aircraft is sampling in fresh smoke, and larger thickness further downwind. Is coating thickness available from the ARCTAS SP2 measurements?**

We do not think that the statement that fresh smoke aerosols are more externally mixed than aged smoke is very controversial, but we do agree we should have provided better references for this claim. For example, Akagi et al. (2012) found that the fraction of BC particles classified as "thickly coated" in the SP2 analysis of a California chaparral fire increased from a value of 0.55 to ~0.80 over four hours of aging. The Colockum Tarp fire sampled during the 2013 Biomass Burning Observation Project (BBOP) also showed a rapid increase in the fraction of thickly coated BC (Sedlacek et al., presentation at the 2014 ASR Science Team Meeting, http://asr.science.energy.gov/meetings/stm/2014/presentations/ASR-talk-20140310.pdf).

Examining the SP2 coating thickness versus smoke plume age would be well beyond the scope of this study. However, to address this comment we have looked at the coating thicknesses in the quasi-Lagrangian sampling of the Lake McKay fire from ARCTAS-B (Alvarado et al., 2010; Cubison et al., 2011) using the median value of the shell/core ratio only for Black Carbon particles with core diameter range of 170-300 nm, which is provided in the ARCTAS data archive. The results are shown in Figure R5 below, and suggest an increase in coating thickness in the first half hour of plume aging.

[Figure]

Figure 5. Median core-shell ratio for coated BC particles with core diameters between 170-300 nm versus estimated smoke age for the early pass of the Lake McKay fire in ARCTAS-B.

However, given that even in fresh smoke about half of the BC particles can be classified as "thickly coated", we agree that our statement in Section 6.2 is too strong. We have thus modified the text (P23, L10-13):

> *In contrast, ARCTAS-B sampled substantial amounts of fresh biomass burning smoke, where about half of the BC would be expected to be externally mixed (e.g., Akagi et al., 2012) and thus have lower absorption per mass of BC than would be calculated by the core-shell assumption.*

**Page 22, line 13: I do not think enough proof is provided regarding mixing state in the observations.**

We agree that the observations do not provide definitive proof that changes in mixing state are responsible for the differences between the three ARCTAS campaign. We have thus softened the language in the revised paper (P23, L17-21)

> *These results do not definitively prove that the difference in the performance of ASP v2.1 for the three ARCTAS campaigns is due to errors in mixing state, but they do suggest the need for further development of the ASP model to allow for time-varying mixing states and to allow the BC size distribution to vary independently of the overall size distribution.*

**Page 22, line 26: Table 7 is presenting the results by mixing rules, not mixing state.**

Agreed, this has been changed to mixing rule in the revised manuscript (P21, L30 and elsewhere).

**Page 24, line 12 and 20: I do not think the model does a reasonable job in absorption, contrary to what the authors state in line 12. Then in line 20 the authors state that there should be improvements in absorption – a contradiction in this paragraph.**

The statement in Line 12 referred to the GMI model, while the statement in line 20 referred to ASP, which performed worse in absorption than GMI, so the statements are not inconsistent. However, we agree that we may have overstated the performance of the GMI model here. We have modified this as follows (P25, L8-11):

> *These results suggest that the GMI model performs the best of the four fixed size distribution schemes evaluated here in simulating submicron aerosol scattering and absorption, and that future refinements to the GMI approach should focus on improvements that, on average, reduce scattering and absorption in the 550/532 nm and 450/470 nm bands.*

**Reviewer 2**

**This paper presents a closure study using in situ observations of aerosol composition and size as well as aerosol optical properties during the ARCTAS campaign to validate five aerosol optical properties modules. During this campaign there was a large abundance of biomass burning aerosol, so the focus of this closure study is on absorption. Four of those modules have in common that they prescribe the aerosol size distribution parameters, so the aerosol mass concentrations are the only inputs. These modules are used in current global models. The fifth optical properties module is used as part of the more detailed aerosol process model ASP and can make use of size distribution information.**

**I regard this paper as a valuable contribution to isolate and quantify errors in optical properties calculations that are part of state-of-the art models. I have some questions and suggestions that would make the presentation of this paper clearer. I recommend this paper for publication after these are addressed.**

**General comments**
**1. Closure studies of optical properties have a long history. I encourage the authors to shift the emphasis in the introduction from describing the importance of LAC to reviewing some of the closure studies in the literature to provide context for the work under review. A few examples are Cai et al., 2011, Highwood et al., 2012, Esteve etal., 2014, Quinn and Coffman, 1998, Sciare et al., 2005, Wex et al., 2002, but there are many more.**

**Cai et al., Journal of Geophysical Research, 116 (2011), D02202**
**Highwood et al., Atmospheric Chemistry and Physics, 12 (2012), 7251–7267**
**Esteve et al., Atmospheric Environment, 89 (2014), 517- 524**
**Quinn and Coffman, Journal of Geophysical Research, 103 (1998), 16575–16596**
**Sciare et al., Atmospheric Chemistry and Physics, 5 (2005), 2253–2265**
**Wex et al., Journal of Geophysical Research, 107 (2002), 8122**

We agree, and thank the author for pointing our attention to the above references. While we still think the discussion of LAC in the introduction is also useful for illustrating the motivation for the work, we have added a discussion of these previous closure studies to our revised paper.

P5, L22 to P6, L5:

> *Previous closure studies have looked at both the scattering and absorption of aerosols measured at surface sites (Sciare et al., 2005; Cai et al., 2011), from research vessels (Quinn and Coffman, 1998) and from aircraft (Wex et al., 2002; Cai et al., 2011; Highwood et al., 2012; Esteve et al., 2014). For example, Quinn and Coffman (1998) found agreement in the submicron scattering calculated via a Mie code using the in situ size distribution and composition measurements and measurements from an integrating nephelometer to within measurement*

*uncertainty, but did not get good agreement for supermicron aerosol. More recently, Highwood et al. (2012) used a Mie code to simulate aerosol scattering (450, 550, and 700 nm) and absorption (567 nm) for several aircraft flights during the EUCAARI-LONGREX campaign, finding agreement within the measurement uncertainties of 30%. Esteve et al. (2014) expanded on this work by using a flexible Mie code assuming homogeneous internally mixed spheres. They found that the agreements between the calculation and measurements of absorption and scattering was within measurement uncertainties for EUCAARI-LONGREX, as in Highwood et al. (2012), but that there was poorer agreement for the VOCALS-Rex campaign, where detailed in situ observations of the aerosol size distribution were not available.*

P6, L14-17:

*This study differs from the closure studies discussed above in that these aerosol optical property modules, which simulate the aerosol as an external mixture of different components each with mixed size distributions, are evaluated directly without the additional constraints provided by the measured aerosol size distributions.*

**2. The description of the models/ aerosol optical property modules is somewhat confusing. I recommend referring consistently to the actual optical property module in each case, since the full chemical transport models are not used for this study. For example, in section 2.2, what is actually evaluated is a portion of the FAST-JX model, correct? Similarly, large parts of section 2.5 seem to be not relevant for this paper, as there are no model runs using the full ASP model, but only the optical properties module is used. If that's indeed the case, I recommend eliminating or substantially shortening section 2.5.1, except for the description about the assumptions how aerosol composition is represented over the different size bins.**

We agree with the reviewer's comment. As noted in the response to Reviewer #1, we have cut most of the extraneous information from Section 2.5.1 as requested. Also, for GMI, GEOS-Chem v9-02, and GC-RT, we are in fact comparing the optical properties that each model feeds into the FAST-JX subroutine for radiative transfer calculations, but as these inputs are different for each model, we think it is simplest to identify them using the name of the host CTMs in which they are used.

**Specific comments**

**1. Replace the phrase "to get" with "to obtain" (e.g. line 5)**

Done – see P4, L5 and elsewhere in the revised text.

**2. page 4, line 9: replace "intensity" with "burden" or similar**

We have changed it to "expense" as recommended by Reviewer #1. See P4, L12 in the revised text.

**3. page 5, line 2: should read "result"**

Done – See P5, L7 in the revised text.

**4. page 6, line 7: suggest to rephrase: "six types of water clouds, three types of ice clouds and ten aerosol types"**

Done – See P6, L31 in the revised text.

**5. page 6, line 12: The list of different components is confusing. Is water-insoluble aerosol everything insoluble except for soot and mineral dust (which are a separate class), and is water-soluble aerosol everything soluble except for sea salt and sulfate? Please clarify.**

We agree that the terminology used in OPAC v3.1 to describe their aerosol components is confusing. Hess et al. (1998) states that the water-insoluble component "consists mostly of soil particles with a certain amount of organic material), while the water-soluble component "consists of various kinds of sulfates, nitrates, and other, also organic, water soluble substances." In practice, most global models use the water-soluble class from OPAC as a starting point for their model of organic carbon (OC) and do not include the water-insoluble class.

We have clarified this in the revised text (see P7, L5-9):

> *The aerosol components included are "water-insoluble" aerosols (primarily soil particles), water-soluble aerosols (primarily organics and other secondary aerosol components, see Hess et al., 1998), soot, two size modes of sea salt, four size modes of mineral dust, and sulfate droplets. A given aerosol is then modeled as an external mixture of these ten aerosol components.*

**6. page 6, line 30: should read "based on Mie theory calculations" (remove "the").**

Done, see P7, L26 in the revised text.

**7. page 7, line 1: How is dependence of refractive indices on relative humidity parameterized?**

As most of our measurements are at very low RH, the dependence of the aerosol optical properties on relative humidity is not very important for our study. However, we agree that we could describe it better in the text. Following OPAC, in most of the fixed-size-distribution models some of the aerosol species are assumed to take on water with increasing RH following prescribed growth curves. The resulting refractive index is then calculated as a volume weighted mixture of the dry aerosol species and water.

We have added this discussion to the text (P7, L28-29):

> *The relative humidity dependent complex indices of refraction (calculated using a simple volume-average mixing rule, see Hess et al., 1998)…*

**8. page 8, line 24: suggest to rephrase: "all particles in a size bin are assumed to have the same composition".**

Done, see P9, L22 in the revised text.

**9. Page 8, line 26: suggest to rephrase: "one bin each for particles smaller than 10 nm or larger than 20 μm."**

Done, see P9, L25-26 in the revised text.

**10. page 8, line 28: This sentence says that the mass fractions of different aerosol components are the same for all the bins. Does this mean that the aerosol is assumed to be fully internally mixed? In other words, it's not only assumed that the particles within one bin have the same composition as stated two lines above, but all particles have the same composition?**

Yes, all size bins are assumed to have the same relative composition. We have edited the text to make this clearer (P9, L27-29):

> *In ASP v2.1, the mass fractions of different aerosol components are assumed to be independent of aerosol size, so the relative aerosol composition is the same in each size bin.*

However, the statement above that all particles within a size bin have the same composition is not true when the external mixture (EXT) mixing rule is used, as in that case we assume that the particles are either pure BC or internal mixtures of the other components. We have thus edited this text to make that clear (P9, L20-23):

> *In this representation, size bin boundaries remain fixed while the mean particle size within the bin is allowed to change with time, and all particles in a size bin are assumed to have the same composition (except when the external mixture (EXT) mixing rule is used, see Section 2.5.2).*

**11. page 10, line 27: Regarding the Maxwell Garnett mixing rule, are there assumptions necessary how many BC inclusions there are, if so, what is assumed in this study?**

As we are dealing with small particles where there is likely only a single BC inclusion, that is what we have assumed in this study for the Maxwell Garnett mixing rule. This is

in contrast to cloud droplets, which could potentially have several BC inclusions. We have clarified this in the text (P10, L19-21):

> ...the Maxwell Garnett (MG) mixing rule, which here assumes that BC is present as a single randomly distributed inclusion within the particle (Maxwell Garnett, 1904);

**12. page 11, line 22: The SP2 instrument only covers a range above 90 nm, but a fraction of BC particles will have smaller cores, which will be missing from the mass concentration, but their absorption will be captured. Can you estimate how big of a problem this is for the closure?**

We agree with the reviewer that this could be a problem. However, this would tend to result in the models under-predicting absorption relative to observations, as a perfect model would have too little BC to represent the measured absorption. However, we found that nearly all models were overestimating absorption, with the exception of ASP v2.1 using the EXT mixing rule at the 470 and 532 nm wavelengths. Thus this error would tend to make the models match observations more closely than they should.

We can make a more quantitative estimate of the effect by estimating the amount of BC mass with diameters below 90 nm. Using the OPAC, GMI, GEOS-Chem v9-02, and GC-RT size distributions for BC gives ~10% of the total BC mass below 90 nm. Thus we expect this missing BC mass would bias the models low in absorption by about 10%. This is less than the SP2 uncertainty (30%), but would be a near constant effect. However, we'd expect it to have less of an influence on our closure study than the differences in the filter corrections for the PSAP.

We have added this discussion to the revised text (P11, L16-22):

> As a rough estimate, we calculate that by not measuring the BC mass below 90 nm, using the measured SP2 mass could bias a perfect absorption model low in absorption by about ~10%. However, as shown in Section 5 and 6, most models studied here overestimate submicron aerosol absorption, so this potential bias would tend to move the models closer to the observations. In addition, this bias is smaller than the potential bias in the absorption measurements due to the filter corrections, as noted in Section 3.3.

**13. page 13, line 13: replace slower with lower.**

Done, see P13, L19 of the revised text.

**14. page 13, line 19: should read 7.75 ?**

This should have read 7.75 μm – we have fixed it in the text (P13, L24).

**15. Section 4.1: Since size distribution measurements are available, can you include some information of how close/how different the observed size distributions were from the assumed distributions?**

The assumptions used for size distributions in most of these the fixed size distribution models are not selected because they are expected to closely match in situ size distribution measurements, but that they are expected to give reasonable aerosol optical properties under a wide variety of condition. Thus we feel that a rigorous comparison of the measured size distributions with the assumed ones for these models would require significant additional work and would be unlikely to provide much benefit.

**16. page 15, line 16: add "volume" to "extinction coefficient"**

Done, see P15, L29 of the revised text.

**17. page 16, line 26: here the authors talk about modes as inputs for ASP, which is in contradiction to the information on page 8, which talked about size bins. Please clarify.**

The issue is that while ASP v2.1 represents the aerosol size distribution using a sectional size distribution, the routines used in ASP v2.1 to initialize this aerosol distribution currently only allow input of an arbitrary number of log-normal aerosol modes which are used to populate the size bins. Thus the combined size distribution has to be represented using log-normal models for input to ASP, but the optical properties are calculated based on the properties of the size bins.

We have clarified this in the revised text (P17, L8-12)

> *While ASP v2.1 represented the aerosol as a sectional size distribution as described in Section 2.5.1, the initialization routines of ASP v2.1 require that the dry aerosol size distributions be input as a sum of lognormal modes which are then used to populate the size bins. Thus the "combined" size distribution described above is fit to three lognormal modes (see Equations 13.18 and 13.20 from Jacobson, 2005).*

**18. page 19, line 2: delete "the shows"**

Done, see P19, L29 of the revised text.

**19. Section 6: the term mixing state is used when it should be mixing rule.**

The reviewer is correct, strictly speaking we are evaluating four different possible mixing rules in ASP, rather than modeling the aerosol mixing state explicitly. However, when we say that further development of ASP should focus on time varying mixing states, here we are referring to the explicit modeling of mixing state. We have clarified this in the abstract (P2, L24-27) and elsewhere in the revised manuscript.

**20. Conclusions: The four fixed-size-distribution models are all rather similar. Can you comment on why the GMI leads to the best results?**

The differences between the four fixed-size distribution models seem to be primarily due to the assumed size distributions and densities of the aerosol components, rather than due to the assumed refractive indices. The simplest comparison is between GMI and OPAC v3.1, as they have the same refractive indices for all species considered here. In terms of absorption, the most significant difference is the assumed density of BC, which is 1.0 g cm$^{-3}$ in OPAC v3.1 and 1.5 g cm$^{-3}$ in GMI, with the assumed size distribution of BC the same in both models. The higher density for BC reduces the volume and cross-sectional area of the BC particles per unit BC mass, reducing the calculated absorption and thus reducing the positive bias for absorption. This improvement is partially offset by the lower density for OC (the other absorbing component). GEOS-Chem v9-02 uses the same refractive index and density for BC and OC as OPAC v3.1, but the larger mode radii and smaller standard deviations of the size distributions result in stronger absorption at 470 nm. In GC-RT, the increase in absorption due to the increase in the imaginary refractive index for BC is offset by the increase in the BC density to 1.8 g cm$^{-3}$.

We have added some of this discussion to the conclusions:

P24, L18-21:
> *The better performance of GMI for absorption seems to be primarily due to the assumption of a larger density for BC (1.5 g cm$^{-3}$) than in OPAC v3.1, as the size distribution and refractive index for BC are the same for these models.*

P25, L2-4:
> *This is likely due to a net reduction in BC absorption due to the increase in the assumed density of BC (1.0 g cm$^{-3}$ in GEOS-Chem v9-02, 1.8 g cm$^{-3}$ in GC-RT) that is partially offset by the increased imaginary refractive index for BC in GC-RT.*

**21. page 24, line 13: What improvements might this be?**

One potential improvement that would reduce both absorption and scattering would be to increase the assumed density of OC to match that used in the other fixed size distribution models. However, this would likely reduce all three wavelengths, not just at 450/470 nm and 550/532 nm. Similarly, the size distribution changes between OPAC and GMI seem to have reduced the scattering more strongly at 700 nm than at 450 or 550 nm. Thus, further investigation and update of the dependence of the complex refractive indices on wavelength might be needed to improve GMI performance.

We have added this discussion to the revised text (P25, L26-31):

> *Increasing the density of OC in GMI to 1.8 g cm$^{-3}$ to match the other fixed-sixe distribution models would likely reduce all three wavelengths equally, and the changes to the fixed size distribution parameters examined here generally*

*resulted in stronger absorption and scattering in all wavelengths, so refining the wavelength dependence of the complex refractive indices used in GMI might be the most promising pathway for significant improvements.*